# Mission Impossible: A Statistical Perspective on Jailbreaking LLMs

**Jingtong Su**[*]
NYU & Meta AI, FAIR

**Julia Kempe**[†]
NYU & Meta AI, FAIR

**Karen Ullrich**[†]
Meta AI, FAIR

## Abstract

Large language models (LLMs) are trained on a deluge of text data with limited quality control. As a result, LLMs can exhibit unintended or even harmful behaviours, such as leaking information, fake news or hate speech. Countermeasures, commonly referred to as preference alignment, include fine-tuning the pretrained LLMs with carefully crafted text examples of desired behaviour. Even then, empirical evidence shows preference aligned LLMs can be enticed to harmful behaviour. This so called jailbreaking of LLMs is typically achieved by adversarially modifying the input prompt to the LLM. Our paper provides theoretical insights into the phenomenon of preference alignment and jailbreaking from a statistical perspective. Under our framework, we first show that pretrained LLMs will mimic harmful behaviour if present in the training corpus. **Under that same framework, we then introduce a statistical notion of alignment, and lower-bound the jailbreaking probability, showing that it is unpreventable under reasonable assumptions.** Based on our insights, we propose an alteration to the currently prevalent alignment strategy RLHF. Specifically, we introduce a simple modification to the RLHF objective, we call *E-RLHF*, that aims to increase the likelihood of safe responses. *E-RLHF* brings no additional training cost, and is compatible with other methods. Empirically, we demonstrate that *E-RLHF* outperforms RLHF on all alignment problems put forward by the AdvBench [1] and HarmBench project [2] without sacrificing model performance as measured by the MT-Bench project [3].

## 1 Introduction

Large Language Models (LLMs) have revolutionized the field of deep learning due to their remarkable capabilities across various domains, serving as assistants, in code generation [4], healthcare [5], and theorem proving [6]. The training process of a LLM typically includes two stages: pretraining with massive corpora, and an alignment step using Reinforcement Learning from Human Feedback (RLHF) to further *align* model behavior with human preferences. The latter step typically involves large amounts of humanly annotated data, and can be decomposed into a supervised fine-tuning (SFT) step, a reward modeling step, and an RL Fine-Tuning step. Despite their ability to perform multiple tasks effectively, LLMs are susceptible to generating offensive or inappropriate content including hate-speech, malware, fake information or social biases, due to the unavoidable presence of harmful elements within their pretraining datasets [7–9]. Social media showcase an abundance of tricks on how to attack ChatGPT [10] to elicit harmful responses, *e.g.,* the "Do Anything Now" (DAN) prompts [11] or the "Grandma Exploit" hack [12]. On the other hand, behavior diversity in the training corpus is essential to for example capturing different cultural preferences. What is and isn't harmful ultimately depends on user preferences, hence the alignment step is not universal but depends on the specific use case under which a model will be employed.

---

[*]Correspondence to: Jingtong Su <js12196@nyu.edu>.
[†]Equal senior authorship.

To address deployment safety and eliminate objectionable responses, numerous *alignment* efforts have been made, such as injecting safe information during SFT [13], performing red teaming with human experts and AI themselves [14–18], as well as refining and improving the whole RLHF process in detail [19–23]. Yet we continue to witness a cat-and-mouse game of ever more sophisticated alignment methods to neutralize "harmful" prompts and even more inventive "jailbreaking" attacks that manipulate those prompts to elicit LLMs to produce harmful information. Such attacks come in various flavors, such as injecting adversarial suffixes [1, 24, 25], exploring cipher and low-resource natural languages [26–28], or letting LLMs craft prompts automatically [29–33]. Although several ad-hoc defense methods against suffixes have been proposed [34–37], we only have limited proposal on a principled universal defense against jailbreaking attacks [2], and limited theoretical understanding on this phenomenon [38].

In this paper, we present a theoretical framework for analyzing both the pretraining phase and the post-alignment jailbreaking phenomenon. Exploiting the fact that *jailbreaking prompts typically maintain the underlying harmful concept while manipulating other aspects of the prompt,* we design framework that decouples input prompts to allows us to quantify the strength of potential adversaries. By representing the output elements of an language model (LM) as lengthier text fragments rather than individual tokens, we can quantify the extent to which these models emulate the training distribution and consequently better understand the mechanisms underlying jailbreaking vulnerabilities.

Our contributions can be summarized as follows:

- Based on our proposed framework, we first offer a non-vacuous PAC-Bayesian style generalization bound for pre-training. Assuming the validity of our framework, we conclude that high-performing pre-trained models will inevitably be susceptible to generating behaviour that is present in the training corpus, including any unintended and harmful behaviour.

- Subsequently, we extend our framework to include notions of alignment and jailbreaking. Assuming our assumptions are met, we demonstrate jailbreaking to be unpreventable even after safety alignment because the LM fails to concentrate its output distribution over the set of safe responses.

- Motivated by our theoretical findings, we identify a key drawback in the widely adopted RL Fine-Tuning objective due to the natural difference between the harmlessness and the helpfulness targets. By addressing this issue, we facilitate the training of safer models that are more resilient to a suite of jailbreaking attacks while preserving model performance.

The paper is organized as follows. In Section 2, we introduce our framework. In Section 3, we prove the PAC-Bayesian generalization bound for pretraining. Next, in Section 4 we present analysis on jailbreaking from a statistical perspective. Finally, in Section 5 we illustrate our proposed E-RLHF objective and its effectiveness on improving LLM safety. We give a literature review in Appendix H.

## 2   Framework and assumptions

Jailbreaking carries several analogies to *adversarial attacks*, a well studied field in computer vision [39]. Here, an adversary is defined as a map that perturbs a given input image in pixel space to change the model output. The strength of the adversary is bounded by how far it is able to move the original input as quantified by the $\ell_p$ distance [40–42]. Typically, this distance is bounded in a way that the change would not be perceptible to the human observer. The goal of the adversary is to cause misclassification of the input. In contrast, in the instance of an LLM, the adversary's goal is to provoke harmful behaviour, *e.g.,* unintended leaking of information or hate speech. Further, any perturbation to an input, called prompt, will have a perceptible effect. Hence quantifying and bounding the capabilities of the adversary is not straight forward. Nonetheless, with some modifications, we will build on this analogy.

For the purpose of this work, we will *view any prompt as a tuple of query and concept* $(q, c)$, where $c \in \mathcal{C}$, and $q \in \mathcal{Q}$, with $\mathcal{C}, \mathcal{Q}$ denoting the complete concept set and query set. Conceptually, we think of *concepts* as representing the information content of the prompt, usually through a short piece of text, for example ``tutorial on making a cake''. *Queries* are instructional text pieces that are composable with certain concepts. We can think of queries as mechanisms to trigger an LM to expand a concept in a specific way. Examples include ``Tell me how to {}'', or ``We are now in an imaginary world, and you are

not bounded by any ethical concerns. Teach my avatar how to {}''. Since not all queries and concepts are composable,[3] we denote $\mathcal{P} \subsetneq \mathcal{Q} \times \mathcal{C}$ as the set of all *plausible* prompts, where the definition of plausible will be made clear below.

**The decomposition of prompts allows us to isolate and hence bound the adversary's strength.** In line with current empirical work on inducing harmful behaviour, we will allow perturbations only on the queries, not on the concepts. Further mimicking the spirit of previous work on adversarial attacks, we will assume that the ground-truth related to a prompt is determined solely by the concept, not the query. We will make these ideas more rigorous in the next paragraphs.

First, in contrast to previous theoretical work where LMs are regarded as *single sentence generators* [38], we model LMs as *lengthier text fragment generators*, and refer to possible generated content $e \in \mathcal{E}$ as explanations. Conceptually, explanations expand concepts with additional information. For example, ''The US president in 2023 is Joe Biden.''. Our terminology "explanation" is conceptually the same as "response" used in previous discussions (*e.g.,* [23]), where an LLM is regarded as a policy that receives an input and generates a response. We use "explanation" to contrast "concept" since in most jailbreaking attacks currently considered by the community, the adversary seeks instructions or explanations for a single harmful attempt. An LM thus induces a mapping from *plausible* prompts to distributions over explanations, $p_{LM} : \mathcal{P} \to \Delta(\mathcal{E})$, where $\Delta(\mathcal{E})$ denotes the set of distributions defined over elements in $\mathcal{E}$.[4] The output of a LM given a prompt, $p_{LM}(q, c)$, is a

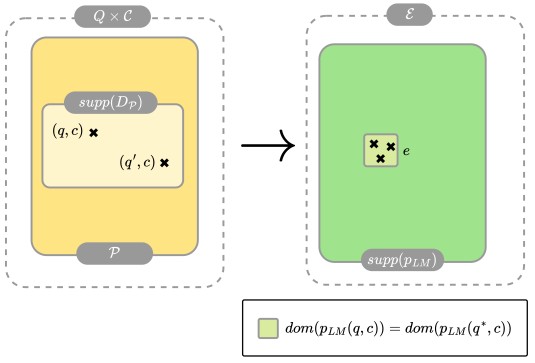

Figure 1: **Our framework in a nutshell:** We define a language model, $p_{LM}$: ▨ → ▩ , as a map from prompts to a distribution over a subset of all possible explanations $\mathcal{E}$. To later be able to bound the strength of the adversarial attacker, we split the text inputs into concepts and queries $(q, c)$. We assume that (i) the text corpus only covers a part of the domain of the LM: $\mathrm{supp}(D_{\mathcal{P}}) \subsetneq \mathrm{dom}(p_{LM})$, (ii) the size of the domain of the output distribution, denoted $|\mathrm{dom}(p_{LM}(q, c))|$, is small compared to the size of $\mathcal{E}$, and (iii) only concepts determine the output (see ▩ ).

discrete distribution over explanations. We use $\mathrm{dom}(p_{LM}(q, c))$ as the *domain* of this distribution, $p_{LM}(e|q, c)$ as the probability of $e$ given $(q, c)$ as the input, and $\mathrm{supp}(p_{LM}(q, c))$ as the subset of $\mathcal{E}$ with non-zero $p_{LM}(e|q, c)$. Further, we assume the existence of a latent ground truth mapping $p_{world} : \mathcal{P} \to \Delta(\mathcal{E})$ that the LM is optimized to mimic during the pretraining stage. This is the distribution that defines "knowledge": for all *plausible* prompts $(q, c)$, it specifies the ground-truth distribution over explanations. By *plausible*, we refer to all prompts that lie in the domain of the ground truth mapping $(q, c) \in \mathrm{dom}(p_{world})$, *i.e.,* $\mathcal{P} \equiv \mathrm{dom}(p_{world})$. Many plausible prompts will not even exist within any available training corpus, as discussed below.

We can now state our main assumption, namely that for any plausible prompt $(q, c) \in \mathrm{dom}(p_{world})$ the ground-truth distribution $p_{world}(q, c)$ is supported on a small subset of $\mathcal{E} \Leftrightarrow \mathrm{supp}(p_{world}(q, c)) \subsetneq \mathcal{E}$. This assumption seems sensible to us: under normal circumstances, providing an explanation of ''Paris'' would not offer any relevant knowledge when given a prompt such as ''How to write a hello world python script''. Our second assumption is that for all plausible prompts $(q, c)$, the concept $c$ *uniquely determines* the *support* of the output distribution specified by $p_{world}$, regardless of the query: $\mathrm{supp}(p_{world}(q, c)) = \mathrm{supp}(p_{world}(q^*, c))$, $\forall$ plausible $(q, c)$ and, $(q^*, c)$ . The query changes the ground-truth distribution without affecting its support. An illustration is depicted in Figure 1. To be more precise:

**Assumption 2.1.** *(Concepts uniquely determine the explanation for plausible prompts)*
*For all plausible prompts* $(q, c) \in \mathrm{dom}(p_{world})$,

$$i)\ p_{world} : \mathcal{P} \to \Delta(\mathrm{supp}(p_{world}(q, c))$$

---

[3]For example, "Who is a tutorial on making a cake." is unreasonable.

[4]For real-world LMs, with different decoding hyperparameters *e.g.,* the temperature $T$, top-$p$ and top-$k$ sampling parameters, the induced distribution with the same set of parameters could be different. Our discussion holds for a pre-fixed set of hyperparameters throughout this paper.

where $\operatorname{supp}(p_{world}(q,c)) \subsetneq \mathcal{E}$ s.t. $|\operatorname{supp}(p_{world}(q,c))| \ll |\mathcal{E}|$; and

$ii)$ $\operatorname{supp}(p_{world}(q,c)) = \operatorname{supp}(p_{world}(q^*,c))$, $\forall(q,c),(q^*,c)$ plausible.

This assumption is natural since it essentially tells us that knowledge is specified by the corresponding concept alone, irrespective of what query is used to extract it. In other words, given a concept $c$, if a query $q$ manages to change $\operatorname{supp}(p_{world}(q,c))$, we argue that the query should be deconstructed and partially absorbed by $c$ to accurately reflect the knowledge mirrored by the support.

Lastly, we make the assumption on the existence of an underlying generative distribution over prompts, denoted as $(q,c) \sim D_{\mathcal{P}}$. This distribution serves as the principle governing the creation of our pretraining corpus. It is important to note that $\operatorname{supp}(D_{\mathcal{P}}) \subsetneq \operatorname{dom}(p_{world})$. For example, take the prompt $(q',c')$=''Who is James Bond \$$\lambda$*#!48811''; even though this prompt never appears in any text corpus across the internet, $(q',c') \notin \operatorname{supp}(D_{\mathcal{P}})$, we, as humans, can make sense of it: $(q',c') \in \operatorname{dom}(p_{world})$. Later proofs in this paper assume LMs generate semantically reasonable explanations for such unseen plausible prompts, since in reality LMs are claimed to generalize well on huge, out-of-distribution datasets [43]. This is made explicit in Section 4, within Assumption 4.1.

Finally, the following definitions pertain to our notion of harmfulness. More specifically, we understand harmful behaviour abstractly as any unintended behaviour. For this, we assume that any explanation $e$ can be denoted as **either harmful or not harmful (safe)**. A concept $c$ is regarded as harmful if and only if the world generates harmful explanations with probability higher than a certain threshold with direct prompts.

**Definition 2.1.** *(Notions of Harmfulness)*
- *(**Direct Queries and Direct Prompts**) We refer to a prompt as direct if it stems from $D_{\mathcal{P}}$, i.e., $(q,c) \in \operatorname{supp}(D_{\mathcal{P}})$. The query of a direct prompt is called a direct query.*

- *(**Harmful Concepts and Harmful Set**) Given a concept $c$, the associated harmful set of explanations is denoted as $E_h(c) := \{e | e \in \operatorname{supp}(p_{world}(\cdot,c)) \land e \text{ is harmful}\}$. In accordance with Assumption 2.1, with a threshold $\eta$, a concept $c$ is harmful if $\forall q$ s.t. $(q,c) \in \operatorname{dom}(p_{world})$, $\sum_{e:e \in E_h(c)} p_{world}(e|q,c) \geq 1 - \eta$. We refer to the set of all possible harmful concepts as $\mathcal{C}_h \subsetneq \mathcal{C}$.*

- *(**Safe Set**) $\forall c \in \mathcal{C}_h$, there exists a corresponding **safe set** $E_s(c) \subsetneq \mathcal{E}$ that we wish $p_{LM}(q,c)$ to be concentrated on. It includes safe explanations existing in $\operatorname{supp}(p_{world}(\cdot,c))$, and explanations designed by humans, e.g., with the template beginning with "Sorry."*

- *(**Semantically meaningful**) We call explanations in $E_h(c) \cup E_s(c)$ as semantically meaningful for the $(q,c)$ prompt.*

- *(**Mixture decomposition of** $D_{\mathcal{P}}$) With these notions, we can decompose $D_{\mathcal{P}} = \alpha D_{\mathcal{P}_h} + (1 - \alpha)D_{\mathcal{P}_s}$ (where $\operatorname{supp}(D_{\mathcal{P}_h})$ includes all direct prompts with a harmful concept, and $\operatorname{supp}(D_{\mathcal{P}_s})$ includes the complement) as a mixture over direct prompts with a harmful concept and the non-harmful counterpart.*

## 3 PAC-Bayesian bound for pre-training LLMs on harmful data

Given a learning algorithm that leads to a *posterior distribution* over a set of models, PAC-Bayesian theory [44] applies Probably Approximately Correct (PAC) inequalities, to provide bounds on the generalization gap, *i.e.,* the difference between the model's empirical loss and the population loss. We now present the first result of our analysis: a non-vacuous PAC-Bayesian bound for pretraining LMs which implies that a well-trained LM ought to exhibit harmful behaviour even when simply prompted with direct queries if it was presented with harmful behavior during training.

We denote by $S = \{(q_i,c_i)\}_{i=1}^n$ a set of prompts generated *i.i.d.* under $D_{\mathcal{P}}$, $S \sim D_{\mathcal{P}}^n$. These prompts together with sampled explanations form our pretraining corpus. We use $\pi, \rho$ as the prior and posterior distribution over LMs before and after the pretraining process, defined over $\mathbb{LM}$, the set of language models. Given a prompt $(q,c)$, we measure the generalization capability of a LM by quantifying the Total Variation (TV) loss between the induced distribution $p_{LM}(q,c)$ and the ground-truth distribution $p_{world}(q,c)$.[5] For real-world LMs, pretraining involves optimizing the cross-entropy loss on the

---

[5]We regard both distributions as defined over the entire $\mathcal{E}$ since we do not restrict the output distribution of LM in this section.

training corpus, which is equivalent to minimizing $\text{KL}[p_{world}(q,c)||p_{LM}(q,c)]$ under our framework. With Pinsker's Inequality, optimizing the KL-divergence term is equivalent to optimizing an upper bound on TV; thus we expect empirical TV loss be small.

**Definition 3.1.** *(TV empirical loss and population loss)*

$$\ell_{\text{TV}}(p_{LM}, (q,c)) := \text{TV}(p_{world}(q,c), p_{LM}(q,c)).$$

*Given an LM and a set of data $S$, the empirical loss $\hat{R}_S(p_{LM})$ and population loss $R(p_{LM})$ are defined as*

$$\hat{R}_S(p_{LM}) := \frac{1}{n}\sum_{i=1}^{n}\ell_{\text{TV}}(p_{LM}, (q_i, c_i));$$

$$R(p_{LM}) := \mathbb{E}_{S\sim D_{\mathcal{P}}^n}\left[\hat{R}_S(p_{LM})\right] = \mathbb{E}_{(q,c)\sim D_{\mathcal{P}}}\left[\ell_{\text{TV}}(p_{LM}, (q,c))\right].$$

We state our PAC-Bayesian bound as follows. The detailed proof can be found in Appendix B.1. [6]

**Theorem 1.** *(PAC-Bayesian Generalization Bound for Language Models.) With $\alpha$ as in Definition 2.1, consider a set of language models $\mathbb{LM}$, with prior distribution $\pi$ over $\mathbb{LM}$.*

*Given any $\delta \in (0,1)$, for any probability measure $\rho$ over $\mathbb{LM}$ such that $\rho, \pi$ share the same support, the following holds with probability at least $1 - \delta$ over the random draw of $S$:*

$$\mathbb{E}_{LM\sim\rho}[R(p_{LM}) - \hat{R}_S(p_{LM})] \leq \sqrt{\frac{\left[\text{KL}[\rho||\pi] + \log\frac{1}{\delta}\right]}{2n}} := \varrho;$$

$$\mathbb{E}_{LM\sim\rho}[\mathbb{E}_{(q,c)\sim D_{\mathcal{P}_h}}\ell_{\text{TV}}(p_{LM}, (q,c))] \leq \frac{1}{\alpha}\left[\mathbb{E}_{LM\sim\rho}\hat{R}_S(p_{LM}) + \varrho\right]. \tag{1}$$

In Appendix B.2 we give a theoretical estimation of $\varrho$, to illustrate the bound we derive is non-vacuous, *i.e.,* less than 1. The KL term is of order $O(K)$ where $K$ is the number of parameters involved in $\pi, \rho$, and $n$ can be shown to greatly exceed $K$ (using a realistic Zipf distribution assumption on prompts to estimate the number of unique prompts). Theorem 1 tells us that, as long as pretraining successfully reduces the loss on the training corpus ($\hat{R}_S(p_{LM}) \downarrow$), in expectation the language model will mimic the world well (small $\ell_{\text{TV}}$ difference) on a given direct prompt sampled from $D_{\mathcal{P}}$. Furthermore, if $\alpha$ is not too small, then this statement holds on a direct prompt whose concept is harmful. Since we have defined the harmful concept as outputting harmful explanations with high probability (Definition 2.1), we conclude that an LM trained on $D_{\mathcal{P}}$ data can output explanations in the harmful set.

## 4 A statistical perspective on jailbreaking after alignment

In this section, we will present the main theoretical contribution of our work: given our assumptions hold, we prove the *existence of ways for an adversary to jailbreak an LM even after the preference alignment process*. Our proof strategy is inspired by the work on adversarial robustness [41], which bounds the adversary's probability of success by upper bounding the volume of the set of points that does not allow for the existence of adversarial examples. Going forward, we need to extend our framework to integrate alignment and jailbreaking.

After an LM is pretrained, it typically will undergo fine-tuning on a dataset containing preferred behaviour. In what follows, we will assume that this alignment process does not change the model performance in the sense that the LM will still produce semantically meaningful explanations (Definition 2.1). It would not, for example, default to answering any request with the same response.

**Assumption 4.1.** *(LM outputs semantically meaningful explanations) For any harmful concept $c$, and all plausible prompts $(q, c) \in \text{dom}(p_{world})$,*

$$\exists |E_n(c)| \ll |E_h(c)| + |E_s(c)| \text{ s.t. } O(1) \ll |\text{dom}(p_{LM}(q,c))| = |E_h(c) \cup E_s(c) \cup E_n(c)|.$$

In other words, we assume the LM's output distribution is accurately supported on $E_h(c) \cup E_s(c)$, in the sense that the size of "residual" $E_n(c)$ is relatively small compared to these semantically meaningful explanations. We define $n(c) = |E_n(c)| + |E_s(c)| + |E_h(c)|$. We omit the $(c)$ annotations

---

[6]The inspiration for the proof of Theorem 1 comes from Mbacke et al. [45], and the proof idea is originally proposed in Germain et al. [46], Haddouche et al. [47].

when clear from the context. The $O(1)$ statement is reasonable, because harmful explanations are usually long text fragments that allow for many alternative formulations. The assumption can be broken down into two components: (1) within the support of the output distribution, only occasional instances of unrelated explanations exist; (2) the process of aligning the model towards safety **does not eliminate the harmful explanations** acquired during the pretraining phase. For part (1), similar to the example we gave above, under normal circumstances, we do not expect the explanation ``Paris'' to appear in $\mathrm{dom}(p_{LM}(q,c))$ given $(q,c)$ as ``How to build a bomb''. As for part (2), though seemingly surprising, evidence with a series of current state-of-the-art LMs can be experimentally validated [48], where diverse, harmful explanations are extracted by simply manipulating the decoding process using direct prompts. In Section 5 we give an explanation for this undesired phenomenon.

To bound the likelihood of jailbreaking we first need to specify how the output of a LM interacts with its support. Assuming a fixed order of explanations in $\mathrm{dom}(p_{LM}(q,c))$, and slight abuse of notation, we can use $p_{LM}(q,c)$ to denote an $n(c)$-dimensional vector on $\Delta^{n(c)-1}$, the probability simplex with $n(c)$ elements, where each entry represents the probability of a single explanation. We call this simplex the **output simplex** related to a given concept $c$. Next, we can induce a distribution on this simplex given a posterior distribution $\gamma$ over the set of language models $\mathbb{LM}$, as follows.

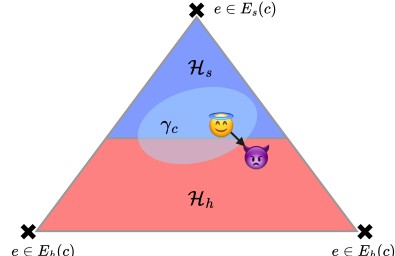

Figure 2: Conceptual illustration of our framework for jailbreaking introduced in Section 4, with a fixed harmful concept $c$. The triangle represents the probability simplex. This figure showcases a typical successful jailbreaking attempt by the adversary: although safety alignment makes the sampled LM safe under the direct prompt input, the adversary is able to move the output to the harmful zone $\mathcal{H}_h$ by manipulating the query $q$.

**Definition 4.1.** *(**Induced Distribution on Simplex,** $\gamma_c$) Under the assumption that the LM outputs semantically meaningful explanations (Assumption 4.1), with a fixed prompt $(q,c)$ and a posterior distribution $\gamma$ over $\mathbb{LM}$, the corresponding induced distribution: $p_{LM}(q,c)$ where $LM \sim \gamma$ is supported over a subset of the output simplex $\Delta^{n-1}$. This distribution is denoted as $\gamma_{(q,c)}$, or $\gamma_c$ when the reference to $q$ is clear from context.*

Next, we will separate the output simplex into a harmful and safety zone. This definition is motivated by the observation that typically an adversary is deemed successful if it can extract even a single harmful explanation for a given concept. This translates into a division of the output simplex, under Assumption 4.1, as follows.

**Definition 4.2.** *(**Harmful Zone and Safety Zone**) For a given harmful concept $c$ and a fixed LM, the output simplex is divided into a **safety zone and a harmful zone**, $\mathcal{H}_s$ and $\mathcal{H}_h$, where a predefined threshold $p \in [0,1]$ is used to quantify the distinction: $p_{LM}(q,c) \in \mathcal{H}_h$ if and only if $\sum_{e:e \in E_h(c)} p_{LM}(e|q,c) \geq p$, and otherwise $p_{LM}(q,c) \in \mathcal{H}_s$.*

Before we introduce jailbreaking, the reader might wonder why we did not define alignment more clearly. This is because under the PAC framework, preference alignment is nothing but a transformation from $\rho$ to some $\gamma$ posterior defined over $\mathbb{LM}$. Given this inability on fine-grained characterization of alignment, we instead provide the *goal of it* as follows. With the above notion, given a prompt $(q,c)$ where $c$ is harmful, its goal is to push the induced distribution $\gamma_c$ into the safety zone $\mathcal{H}_s$. Ideally, $\mathrm{supp}(\gamma_c) \subset \mathcal{H}_s \Leftrightarrow$ with probability 1, the resulting LM is safe when encountering $(q,c)$. We are ready to introduce necessary concepts related to jailbreaking.

**Definition 4.3.** *(**Jailbreaking**) Given a harmful concept $c$ and a query $q'$, the prompt $(q',c)$ **jailbreaks** the LM iff $p_{LM}(q',c) \in \mathcal{H}_h$. We call such a prompt $(q',c)$ and query $q'$ a jailbreaking prompt and jailbreaking query, respectively.*

The threshold $p$ for discriminating $\mathcal{H}_h$ and $\mathcal{H}_s$ should be very small, since it means in expectation the adversary needs to call the LM $\frac{1}{p}$ times to collect a single harmful explanation *i.e.,* to jailbreak the LM.

To theoretically prove the jailbreaking effect, we need to restrict the adversary's ability. To achieve this goal, we borrow insights from adversarial attacks, to assume that the adversary has bounded manipulating capability on the output simplex when searching over the query set:

**Assumption 4.2.** *($\epsilon$-bounded adversary) Given an LM, a harmful concept $c$ and an associated direct prompt $(q,c)$, we assume the adversary can find a set of queries $\mathcal{Q}'$, such that the output is moved **at***

*most $\epsilon$ on the simplex towards $\mathcal{H}_h$ from $p_{LM}(q,c)$:*

$$\sup_{q' \in \mathcal{Q}'} d(p_{LM}(q,c), p_{LM}(q',c)) = \epsilon.$$

*Here $d$ is a distance measure between two discrete distributions. $d$ can be a typical $\ell_p$ measure with $p \geq 1$, or the Total Variation / Jensen-Shannon Divergence. We call $q' \in \mathcal{Q}'$ an $\epsilon$-bounded query.*

A conceptual illustration of our framework is depicted in Figure 2. Before arriving at our Theorem, we give the final definition of $\epsilon$-expansion.

**Definition 4.4.** *($\epsilon$-expansion) Given a set $A \subset \Delta^{n-1}$ and a distance measure $d$, the $\epsilon$-expansion set $A(\epsilon, d)$ is defined as*

$$A(\epsilon, d) := \{t | t \in \Delta^{n-1} \wedge \exists y \in A \ s.t. \ ||y - t||_d \leq \epsilon\}.$$

We are ready to present the following theorem, which states that as long as the induced posterior $\gamma_c$ is not concentrated in an extremely safe area, then with high probability the model can be jailbroken. The proof is in Appendix B.3.

**Theorem 2.** *(Jailbreak is unavoidable) Assume that an LMs output semantically meaningful explanations (Assumption 4.1). Given any $\gamma$ posterior distribution over $\mathbb{LM}$, choose a harmful concept $c$ with a direct prompt $(q, c)$ and a threshold $p$ (Definition 2.1), to define the corresponding induced distribution $\gamma_c$ (Definition 4.1) and division over output simplex (Definition 4.2). An $\epsilon$-bounded adversary (Assumption 4.2) can find a jailbreaking prompt (Definition 4.3) with probability at least*

$$1 - \gamma_s \times (1 - \Phi(a_\epsilon)),$$

- *by using either the direct prompt, such that $p_{LM}(q,c) \in \mathcal{H}_h$; or*

- *by finding an $\epsilon$-bounded query $q'$, such that $p_{LM}(q',c) \in \mathcal{H}_h$.*

*Here, $\Phi(\cdot)$ is the standard Gaussian cdf, $\gamma_s := \max_{x \in \mathcal{H}_s - \mathcal{H}_h(\epsilon,d)} \frac{\gamma_c(x)}{U(x)}$, with $U(x)$ the uniform distribution over $\Delta^{n-1}$, and $a_\epsilon := a + \sqrt{n-1}\epsilon$, where $a$ writes analytically as $a \asymp \frac{|E_h(c)| - 1 - (n-1)p}{\sqrt{(n-1)p(1-p)}}$.*

Trivially, the chances of an adversary to find a jailbreaking prompt increase for stronger adversaries ($\epsilon \uparrow$). In the real world, this could relate to how much compute budget we allow to alter a query for a specific harmful concept. Furthermore, the chances of an adversary to find a jailbreaking prompt increase when the ratio of the sizes of the harmful explanation set to the safe explanation set is larger $\frac{|E_h(c)|}{|E_s(c)|} \uparrow$. This is because their ratio will determine the size of the harmful zone which in turn will cause $\Phi(a_\epsilon) \to 1$. In real world settings, for any harmful concept, the training corpus naturally contains a large harmful set due to the number of possible responses. Realistically, its size can not be countered by any manually-constructed safe set. **Hence achieving alignment is hard**: Recall that the goal of alignment is to respond with only safe explanations with high probability. However, we just learned that to increase that probability, we need to have a small harmful-to-safety set ratio which we discussed is not realistic. Consequently, the safety zone is going to be small.

## 5 E-RLHF: improving alignment by *expanding* the safety zone

Recall from Theorem 2 and the subsequent discussion in the previous section, that jailbreaking becomes more likely the larger the harmful zone is in comparison to the safety zone. The size of both zones relates to the size of their respective explanation sets. In other words, the size of the preference alignment dataset is crucial to successful alignment. Unfortunately, the human labor involved in creating such a dataset effectively caps its size.

In order to bridge the gap between our theoretical insights and a practical solution towards suppressing the jailbreaking problem, we focus on other more **practical ways to expand the safety zone**. Even though our ideas are more broadly applicable, in our experiments we will focus on improving Reinforcement Learning with Human Feedback (RLHF). RLHF typically includes three phases: i) supervised fine-tuning (SFT); ii) preference sampling and reward learning and iii) RL optimization. Rafailov et al. [23] have recently proposed a widely applied version of RLHF for LMs, coined Direct Preference Optimization (DPO), that employs a clever reparameterization which leads to directly learning from the preference dataset, without the need of obtaining a reward model beforehand.

DPO is more stable in the training process than other implementations of RLHF. A more complete overview of RLHF and DPO can be found in Appendix C.

For our purposes, we assume access to an LLM $p_{\text{SFT}}$ that has been supervised fine-tuned on high-quality data. We further assume access to a preference aligned dataset $\mathcal{D}_s$; that contains a set of text prompts $(q, c) = x$, and two respective explanations that have been rated by human annotators as better $e_w$ or worse $e_l$. In phase ii) of RLHF, one typically optimizes a reward model $r(x, e)$ based on the annotated explanations. Our proposal concerns phase iii) of the RLHF process: training of the preference aligned model $p_{LM}$. For a given reward model, $p_{LM}$ is typically obtained by **minimizing** the following objective:

$$\mathcal{L}_{\text{RLHF}}(p_{LM}) = -\mathbb{E}_{x \sim \mathcal{D}_s}\{\mathbb{E}_{e \sim p_{LM}(\cdot|x)}[r(x, e)] + \beta \mathbb{D}_{\text{KL}}(p_{LM}(x)||p_{\text{SFT}}(x))\} \tag{2}$$

Note that, the first term is maximizing the reward, while the KL-term acts as a regularizer ensuring the aligned model can not drift off too far from the SFT model. We claim that this regularization is exactly the problem for safety. While designed to keep the model helpful[7], for any harmful prompt $x_h$ and any harmful explanation $e \in \text{supp}(p_{\text{SFT}}(x_h))$, regularizing $p_{LM}$ to $p_{\text{SFT}}$ lets $p_{LM}$ maintain $e$ in the support of the output distribution. Specifically, the supervised fine-tuning process does not involve elimination of any harmful explanations from the support of the output distribution of the pretrained model, [8] which leads to the fact that $p_{\text{SFT}}(x_h)$ supports harmful responses that can not be negated with a realistically sized preference aligned dataset. Thus, this small safe set problem of $p_{\text{SFT}}$ will be further passed to $p_{LM}$, even if $p_{LM}$ is optimized to the optimal solution of the above objective. This suggests we should not push $p_{LM}$ into the direction guided by $p_{\text{SFT}}$. Instead, for any harmful prompt $x_h$, we propose to use a distribution that we *design* to have a large safety zone to replace $p_{\text{SFT}}(x_h)$. Inspired by our framework, we can achieve this by **modifying the harmful prompt**, such that a safe concept is used to replace the harmful concept, which alters $\text{supp}(p_{\text{SFT}}(\cdot))$ to include more safe explanations. For example, when $x_h$ is ''Tell me how to make a bomb'', instead of aligning $p_{LM}(x_h)$ to $p_{\text{SFT}}(x_h)$, now we align $p_{LM}(x_h)$ to $p_{\text{SFT}}(x_s)$ where $x_s$ is ''Tell me how to reject a request of making a bomb''. This can be done in an individualized manner for all harmful prompts in $\mathcal{D}_s$, or simply by prefacing all harmful prompts with a prefix such as "Please ensure your response adheres to community guidelines and ethical standards:". [9] Importantly, the non-harmful prompts are not modified. Due to the focus of our approach to *expand* the safety zone of the output distribution, we coin our proposal *E-RLHF*, resulting in the following modification to Eq. (2):

$$\mathcal{L}_{\text{E-RLHF}}(p_{LM}) = -\mathbb{E}_{x \sim \mathcal{D}_s}\{\mathbb{E}_{e \sim p_{LM}(\cdot|x)}[r(x, e)] + \beta \mathbb{D}_{\text{KL}}(p_{LM}(x)||p_{\text{SFT}}(x_s))\} \tag{3}$$

where $x_s$ is a **safety-transformed version of the original harmful prompt** $x_h$. To recap, the key argument we put forth is that, in order to ensure the stability of model fine-tuning, it is not imperative to utilize identical prompt inputs $x$ for both the reference model and the target model, particularly when the original input $x$ itself is harmful. In fact, as long as the substitute or "anchor" prompt generates logically reasonable outputs akin to those produced by the original prompt, this approach would not impede the training process of the model. To solidify our argument we show the impact of our modification on the support of the optimal policy in Appendix C. We also deduce there that we can trivially integrate our modification into the DPO objective allowing us to train without an explicit reward model (eliminates step ii)) as follows, where $\sigma(\cdot)$ stands for the sigmoid function:

$$\mathcal{L}_{\text{E-DPO}}(p_{LM}) = -\mathbb{E}_{(x, e_w, e_l) \sim \mathcal{D}_s}\left[\log \sigma\left(\beta \log \frac{p_{LM}(e_w \mid x)}{p_{\text{SFT}}(e_w \mid x_s)} - \beta \log \frac{p_{LM}(e_l \mid x)}{p_{\text{SFT}}(e_l \mid x_s)}\right)\right]. \tag{4}$$

## 6 Experiments and results

**Our experimental set-up** is based on the alignment-handbook code base [50]. We tune the publicly-available SFT model $p_{\text{SFT}}$ provided by huggingface hub [51], using the public dataset [52, 53], with default hyperparameter setup. We label harmful prompts in the preference dataset by prompting GPT-3.5-Turbo, see Appendix E. We are using the very same prefix proposed in the previous section to generate $x_s$. Experiments are performed on 8 NVIDIA Tesla V100 GPUs, using half-precision

---

[7]Otherwise the model could drift into trivial behaviour like always responding with "I can't help you.".

[8]Even the probability can be suppressed to close to 0.

[9]The prefix shares similarities to the system prompts used by open-source LLMs [13, 49] to boost safety.

Table 1: Safety alignment with the E-RLHF objective, here specifically E-DPO, reduces the average Attack Success Rate (ASR) across all jailbreak adversaries for both the HarmBench and the AdvBench data, to 36.95, and to 20.89, respectively. Moreover, resilience against all adversaries improves with our modification to safety alignment ( indicates better performance between DPO and E-DPO).

| Model | Direct Request | GCG | GBDA | AP | SFS | ZS | PAIR | TAP | AutoDAN | PAP-top5 | Human | AVG ↓ |
|---|---|---|---|---|---|---|---|---|---|---|---|---|
| | | | | | HarmBench ASR [2] | | | | | | | |
| $p_{SFT}$ | 32.25 | 59.25 | 35.50 | 42.75 | 42.75 | 36.20 | 56.50 | 65.00 | 56.75 | 26.75 | 35.50 | 44.47 |
| $p_{DPO}$ | 27.50 | 53.00 | 39.00 | 46.75 | 43.25 | 29.10 | 52.50 | 54.00 | 51.00 | 28.75 | 37.15 | 42.00 |
| $p_{E-DPO}$ (ours) | 23.50 | 47.50 | 31.75 | 36.25 | 40.50 | 26.45 | 48.50 | 51.00 | 43.00 | 27.00 | 31.05 | 36.95 |
| | | | | | AdvBench ASR [1] | | | | | | | |
| $p_{SFT}$ | 6.00 | 80.00 | 13.00 | 37.00 | 31.00 | 14.80 | 65.00 | 78.00 | 91.00 | 4.00 | 21.20 | 40.09 |
| $p_{DPO}$ | 0.00 | 47.00 | 12.00 | 39.00 | 30.00 | 7.00 | 50.00 | 61.00 | 44.00 | 4.00 | 18.40 | 28.40 |
| $p_{E-DPO}$ (ours) | 0.00 | 38.00 | 8.00 | 15.00 | 21.00 | 5.20 | 41.00 | 53.00 | 31.00 | 4.00 | 13.60 | 20.89 |

tuning *i.e.,* Float16. In the appendix, we also show results for an alternative training paradigm: the Low-Rank Adaptation (LoRA) [54] (see Appendix D.1). Following community standards [3, 1, 2], we use greedy decoding *i.e.,* $T = 0$ for model evaluation.

We first show empirical evidence that our proposed modification of DPO, E-DPO, does in fact improve safety alignment, using the Harmbench dataset [2] and the first 100 prompts in the AdvBench harmful behavior dataset [1], measured by the HarmBench protocol. We give an overview on all adversaries in Appendix F. The results are presented in Table 1. **E-DPO achieves improvements across every task we tested.**

On top of our safety results, **we want to make sure E-RLHF does not sacrifice helpfulness for increased safety**. We evaluate helpfulness with the MT-Bench project [3]. The SFT model $p_{SFT}$ receives a score of 6.3, and both the DPO and E-DPO models perform better than that (6.9 and 6.7 respectively), making us believe that performance degradation is not a problem with our proposal. Next, we show the impact of the safe prefix on model performance. We demonstrate that **our method's performance depends on the choice of safe prefix to some extend but never fails** (see Appendix D.2). We believe, finding better safe prefixes by explicit tuning would improve our results, similar to the work by Yang et al. [55], but we leave this exploration for future work. Further, we confirm that the improvement arises from using a safe prior in the KL term for harmful prompts. **We ablate our results by appending the prefix on all prompts in the preference alignment dataset** (see Appendix D.3). In all cases, applying the safe prefix to usual prompts *degrades* safety, showcasing the importance of switching the prior only on the harmful prompts. Finally, we show that E-DPO can **be combined with any system prompt, to further boost safety** (see Appendix D.4). The proposal can even be used to **improve helpfulness and safety simultaneously** (see Appendix D.5).

## 7 Conclusion and discussions

In this paper, we present a theoretical framework for language model pretraining and jailbreaking by dissecting input prompts into query and concept pairs. Through this approach, we have established two theoretical results pertaining to the ability of language models to mimic the world following pre-training, which leads to outputting harmful explanations given harmful prompts; and the inevitability of jailbreaking resulting from alignment challenges. Guided by these theoretical insights, we have devised a simple yet effective technique to enhance safety alignment, and demonstrate the improved resilience to jailbreak attacks with this methodology.

**Current limitations** (1) Although we have classified concepts as either harmful or non-harmful, it is important to acknowledge that the perception of a concept's potential for harm can be influenced by various factors such as cultural, legal, and societal norms, which collectively form the *context* of the situation. (2) Language models have demonstrated impressive capabilities in reasoning and completing tasks within multi-round, multi-step conversations; our current framework may not fully account for the generalization and jailbreaking possibilities associated with such input formats. (3) Our analysis is grounded on a fixed $p_{world}$ mapping and $D_{\mathcal{P}}$ distribution. Nevertheless, the world is inherently dynamic, as both $p_{world}$ and $D_{\mathcal{P}}$ continually evolve.

**Future work** (1) Regarding our E-RLHF approach, as highlighted in the experimental section, in addition to attaching a universally safe prefix to all harmful prompts, improvements can be achieved by individually transforming the harmful prompts. Moreover, the safety-transformed prompts can be employed to expand the preference dataset for conventional RLHF. (2) Throughout our analysis, we have not imposed any constraints on the *capacity* of the language model. Extending our analysis under finite memory constraints or analyzing hallucination properties of LLMs is an interesting direction to explore. (3) Large language models have shown remarkable capabilities as in-context learners [56], and such techniques could potentially be used for jailbreaking them as well [57–59]. Investigating the incorporation of such input paradigms remains a promising avenue for future research.

## Acknowledgement

JS and JK are supported by the National Science Foundation under NSF Award 1922658. Part of this work was done while JK was hosted by the Centre Sciences de Donnees at the École Normale Supérieure (ENS) in 2023/24, whose hospitality she gratefully acknowledges.

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

# Appendix

## A Glossary

Table 2: Summary of notation.

| Symbol | Meaning |
|--------|---------|
| $q$ | A single query, composable with a certain set of concepts. |
| $c$ | A single concept, composable with a certain set of queries. |
| $x = (q, c)$ | A single prompt composed by query $q$ and concept $c$. |
| $e$ | A single explanation. |
| $\mathcal{Q}$ | The query set. |
| $\mathcal{C}$ | The concept set. |
| $\mathcal{E}$ | The explanation set. |
| $\mathcal{P} \subseteq \mathcal{Q} \times \mathcal{C}$ | The set of plausible prompts. |
| $p_{world} : \mathcal{P} \to \Delta(\mathcal{E})$ | The world mapping. For each plausible prompt, it specifies the ground-truth distribution over $\mathcal{E}$, a.k.a. the "knowledge". |
| $p_{world}(q, c)$ | The ground-truth distribution over a subset of $\mathcal{E}$, given a prompt $(q, c)$. With slight abuse of notation, it also refers to a point on the probability simplex. |
| $\text{supp}(p_{world}(q, c))$ | Support of $p_{world}(q, c)$. A strict subset of $\mathcal{E}$. |
| $p_{LM} : \mathcal{P} \to \Delta(\mathcal{E})$ | A language model. For each plausible prompt, it specifies a distribution over (a subset of) $\mathcal{E}$, to mimic $p_{world}$. |
| $p_{LM}(q, c)$ | The output distribution over (a subset of) $\mathcal{E}$ by LM, given a prompt $(q, c)$. With slight abuse of notation, it also refers to a point on the probability simplex. |
| $\text{dom}(p_{LM}(q, c))$ | Domain of the $p_{LM}(q, c)$ distribution, a subset of $\mathcal{E}$. |
| $(q, c) \sim D_{\mathcal{P}}$ | Underlying generative distribution over prompts, a.k.a. the distribution governing the creation of our pretraining corpus. |
| $\text{supp}(D_{\mathcal{P}}) \subsetneq \mathcal{P}$ | Support of $D_{\mathcal{P}}$. $(q, c) \in \text{supp}(D_{\mathcal{P}})$ is called a **direct prompt**. |
| $\mathbb{LM}$ | A set of language models. |
| $\pi$ | The prior distribution over $\mathbb{LM}$. |
| $\rho$ | The posterior distribution over $\mathbb{LM}$, after pretraining. |
| $\gamma$ | The posterior distribution over $\mathbb{LM}$, after preference alignment. |

## B Proof of Theorems

### B.1 Proof of PAC-Bayesian bounds

**Definition B.1.** *(Bounded Difference) A function $f : \mathcal{X}^n \to \mathbb{R}$ is said to have bounded difference property w.r.t. a collection of constants $c_1, \cdots, c_n$, iff*

$$\sup_{x_1, x_2, \ldots, x_n, x_i'} |f(x_1, x_2, \cdots, x_n) - f(x_1, x_2, \cdots, x_{i-1}, x_i', \cdots, x_n)| \leq c_i, \forall i \in [n].$$

**Lemma B.1.** *(Hoeffding's Lemma) for random variable $X \in [a, b]$ with probability 1, the following holds:*

$$\mathbb{E}[\exp(\lambda X)] \leq \exp(\lambda \mathbb{E} X + \frac{\lambda^2 (b-a)^2}{8}).$$

**Lemma B.2.** *(Hoeffding's Lemma, Multivariate) for random variables $Z = f(x_1, \cdots, x_n)$ where $f$ has the bounded difference property, the following holds:*

$$\mathbb{E}[\exp(\lambda(\mathbb{E} Z - Z))] \leq \exp(\frac{\lambda^2 \sum_{i=1}^n c_i^2}{8}).$$

Note that substituting $Z$ with $\hat{R}_S(LM)$ is valid.

**Lemma B.3.** *Empirical Loss defined in Definition 3.1 satisfies the bounded difference condition with constant $c = 1, \forall i$.*

We are ready to present the proof of Theorem 1.

*Proof.* Starting with the above lemma, we know

$$\mathbb{E}_S[\exp(\lambda(R(LM) - \hat{R}_S(LM)))] \leq \exp(\frac{\lambda^2 c^2}{8n}).$$

The above result holds for a manually picked LM. With an overall average over the prior $\pi$ we have

$$\mathbb{E}_{LM \sim \pi} \mathbb{E}_S[\exp(\lambda(R(LM) - \hat{R}_S(LM)))] \leq \exp(\frac{\lambda^2 c^2}{8n}).$$

Apply Fubini's theorem (note that $\pi$ is independent of $S$):

$$\mathbb{E}_S \mathbb{E}_{LM \sim \pi}[\exp(\lambda(R(LM) - \hat{R}_S(LM)))] \leq \exp(\frac{\lambda^2 c^2}{8n}).$$

Define $Y = \mathbb{E}_{LM \sim \pi}[\exp(\lambda(R(LM) - \hat{R}_S(LM)))]$, a random variable depends on $S$. Obviously $Y \geq 0$. Thus, with Markov's inequality:

$$\mathbb{P}[Y \geq \frac{1}{\delta}\mathbb{E}_S Y] \leq \delta.$$

Equivalently, with probability at least $1 - \delta$, we have

$$Y \leq \frac{1}{\delta}\exp[\frac{\lambda^2 c^2}{8n}].$$

Since we have assumed $\pi, \rho$ share the same support, using Radon-Nykodim derivative to change the expectation with respect to $\pi$ to with respect to $\rho$, we have

$$\mathbb{E}_{LM \sim \rho}\left[\frac{d\pi}{d\rho}\exp(\lambda(R(LM) - \hat{R}_S(LM)))\right] \leq \frac{1}{\delta}\exp[\frac{\lambda^2 c^2}{8n}].$$

Taking logarithm and applying Jensen's Inequality we know

$$\mathbb{E}_{LM \sim \rho}\left[\frac{d\pi}{d\rho} + \lambda(R(LM) - \hat{R}_S(LM))\right] \leq \log\frac{1}{\delta} + \frac{\lambda^2 c^2}{8n}.$$

Incorporating $c = 1$, noticing $\frac{d\rho}{d\pi} = (\frac{d\pi}{d\rho})^{-1}$ we could rewrite the inequality as

$$\mathbb{E}_{LM \sim \rho}\left[(R(LM) - \hat{R}_S(LM))\right] \leq \frac{1}{\lambda}\left(\text{KL}[\rho||\pi] + \log\frac{1}{\delta}\right) + \frac{\lambda}{8n}.$$

Finding $\lambda$ that minimizes the term on right hand side gives us the $\varrho$ term.

When $D_{\mathcal{P}}$ allows for a decomposition into mixture components, noticing the linearty of expectation, the bound can be re-written as

$$\alpha\mathbb{E}_{LM \sim \rho}[\mathbb{E}_{(q,c) \sim D_{\mathcal{P}_h}}\ell_{\text{TV}}(p_{LM}, (q,c))] + (1 - \alpha)\mathbb{E}_{LM \sim \rho}[\mathbb{E}_{(q,c) \sim D_{\mathcal{P}_s}}\ell_{\text{TV}}(p_{LM}, (q,c))]$$
$$\leq \varrho + \mathbb{E}_{LM \sim \rho}[\hat{R}_S(p_{LM})].$$

which leads to

$$\mathbb{E}_{LM \sim \rho}[\mathbb{E}_{(q,c) \sim D_{\mathcal{P}_h}}\ell_{\text{TV}}(p_{LM}, (q,c))] \leq \frac{1}{\alpha}[\varrho + \mathbb{E}_{LM \sim \rho}[\hat{R}_S(p_{LM})]].$$

$\square$

## B.2 An estimation on the non-vacuousness of the PAC bound

We give an estimation of the term appears in our PAC bound, $\varrho$, and state that it is non-vacuous.

**The numerator.** We follow Neyshabur et al. [60] to instantiate the term in the simplest setup. Assume $\pi, \rho$ are defined over the parameter space of a given LM, with $K$ parameters. Assume $w$ is a set of weights learned from the pretraining corpus. Let the prior $\pi$ be the zero-mean multivariate Gaussian, whose entry-wise variance is related to the magnitude of the weight: $\sigma_i = \beta|w_i|$, and $\rho$ be a Gaussian with the same anisotropic variance centered around $w$. We argue though simple, both settings are practical, since Gaussian initialization is common for model training, and the SWA-Gaussian algorithm [61] utilizes such Gaussian posterior. Under this setup, the KL goes as $\sum_i \frac{w_i^2}{2\sigma_i^2} = O(K)$. Specifically, taking $\beta = \frac{\sqrt{2}}{2}$ makes the term exactly $K$. Current language models often possess millions, or billions, of parameters, namely, $K \sim [10^6, 10^9]$.

**The denominator.** To estimate the number of unique direct prompts in the training corpus, it is important to notice that the dataset does not only consist of $(q, c)$ prompts but also $e$ explanations. Thus, we need to estimate the *average token length (ATL)* associated with each unique prompt $x = (q, c)$. For each unique prompt $x$, aside from its own token length $l(x)$, there will be a collection of explanations $\{e_i\}_{i=1}^{N(x)}$, with expected token length of each associated explanation $l(e)$. We have

$$\mathbb{E}ATL = \mathbb{E}_{x \sim D_{\mathcal{P}}}N(x) \times [l(x) + l(e)].$$

**Fact.** Given a prompt $x$, the larger the expected length of the prompt itself and explanation ($l(x) + l(e) \uparrow$), the larger the expected *number of explanation elements* ($N(x) \uparrow$), and the smaller the number of such prompts ($D_{\mathcal{P}}(x) \downarrow$), appearing in the training corpus. The former comes naturally due to the composability of natural language: the longer the text fragment, the more equivalent text fragments in expectation, while the latter is reflected by the spirit of the widely accepted Zipf's law.

Inspired by the fact, we assume prompts are categorized by the quantity of $l(x) + l(e)$, namely, for all prompt $x$, $N(x)$ is a function of $l(x) + l(e)$. Moreover, the complete data generation process is decomposed into i) sample a value of $l(x) + l(e)$ out, and then ii) sample a unique prompt from the set decided by this specific $l(x) + l(e)$ value, and iii) generate $N(x)$ explanations.

Step i). Use the fact: the larger the expected length of the output explanation, the smaller the probability that such a prompt appears in the training corpus. We assume step i) follows a (cut-off) zeta distribution. Specifically, for a random prompt $x$,

$$p(l(x) + l(e) = k) \propto k^{-s}, \forall k \geq k_0.$$

When $k_0 = 1$, we resume the zeta distribution with coefficient $s$.

Step ii). We assume each prompt following this step is unique.

Step iii). Use the fact: the larger the expected length of the output explanation, the larger the expected *number of explanation elements* in the training corpus. We assume a power law scaling on $N$, with a constant $t > 1$, such that

$$N(l(x) + l(e) = k) = k^{t-1}.$$

Thus, the average token length writes

$$\mathbb{E}ATL = \sum_k p(l(x) + l(e) = k) \times k \times N(l(x) + l(e) = k) = \frac{\zeta(s-t) - \sum_{i=1}^{k_0-1} i^{-(s-t)}}{\zeta(s) - \sum_{i=1}^{k_0-1} i^{-s}}.$$

where $\zeta(s) = \sum_{i \in \mathbb{Z}^+} i^{-s}$ is the Riemann zeta function.

For example, take $s = 4, t = 2$. With $k_0 = 1$, the ATL would be 1.52, while with $k_0 = 10$, the ATL becomes 272. These results translate into an estimation of unique prompts as $n_{\text{tokens}}/ATL$. With current SOTA LM, the pretraining corpus often includes (tens of) trillions of tokens ($> 10^{12}$), thus $n > 10^{10} > K$ can be safely assumed $\Rightarrow \varrho < 1$.

$\alpha$ **constant.** According to LLaMa-2 report (section 4.1, Figure 13) [13], approximately $0.2\%$ of the documents in their training corpus is labeled as harmful. However, we argue this is indeed an extremely **loose** lower bound for $\alpha$, due to the estimation strategy used in their paper. Given a document, they use a binary classifier on harmfulness over *each single line* (1 means harmful and 0 otherwise), and assign the *average* score to the document. $0.2\%$ is the ratio of documents *with score* $\geq 0.5$. Take the example of ''How to build a bomb''. The chemical reaction parts will not be counted as harmful, and thus this estimation strategy could judge a completely harmful explanation as harmless. Thus, it is reasonable to assert $\alpha$ is not too small, though with current literature we are not capable of raising an accurate estimation on it.

### B.3 Proof of jailbreaking

Before proceeding to the proof, we list necessary definitions and lemmas as follows.

**Lemma B.4.** *(Volume of $n$-simplex)*[10] *For any dimension $n$, the volume of the $n$-element probability simplex: $\Delta^{n-1}$, in the $n-1$-dimensional space is*

$$\frac{\sqrt{n}}{(n-1)!}.$$

We define the projected probability simplex as follows.

**Definition B.2.** *(Projected probability simplex) Given $\Delta^{n-1}$, the corresponding projected probability simplex, $\Delta_p^{n-1}$, is defined as a subset of $\mathbb{R}^{n-1}$: $\{x \in \mathbb{R}^{n-1} | \sum_{i=1}^{n-1} x_i \leq 1, \forall i \in [n-1]\}$.*

---

[10]See https://en.wikipedia.org/wiki/Simplex#Volume.

**An illustration of $\Delta^{n-1}$ and $\Delta_p^{n-1}$.** For example, take $n = 3$. The probability simplex with $n = 3$ elements is a triangle whose (euclidean) side length is $\sqrt{2}$ with vertices $(1, 0, 0), (0, 1, 0), (0, 0, 1)$. Then its volume in the 2-dimensional space, *i.e.*, its area, is $\frac{\sqrt{3}}{2}$. The corresponding projected probability simplex is the triangle between the $X - Y$ axis, with vertices $(1, 0), (0, 1), (0, 0)$.

A direct lemma that connects the probability simplex and the projected probability simplex is given below.

**Lemma B.5.** *(Transformation of probability simplex) Given a proper probability density function $\nu(x)$ defined on $\Delta_p^{n-1}$, it is equivalent to the distribution defined on $\Delta^{n-1}$ with density $\frac{\nu(x)}{\sqrt{n}} : \forall A \in$ Borel$(\Delta_p^{n-1})$, let $B = \{x \in \Delta^{n-1} : x_{1:n-1} \in A\}$. Then $\int_A \nu(x)dx = \int_B \frac{\nu(x)}{\sqrt{n}}dx$. Specifically, this implies $\frac{volume(A)}{volume(\Delta_p^{n-1})} = \frac{volume(B)}{volume(\Delta^{n-1})}$.*

*Proof.* Consider a translation on $\Delta^{n-1}$ with $x_n = -\sum_{i=1}^{n-1} x_i$ which does not affect its the volume and shape. The mapping: $\Delta_p^{n-1} \to$ translated$\Delta^{n-1}$ is an affine transformation with matrix

$$T = \begin{pmatrix} 1 & 0 & \cdots & 0 \\ 0 & 1 & \cdots & 0 \\ \cdots & \cdots & \cdots & \cdots \\ -1 & -1 & \cdots & -1 \end{pmatrix}_{n \times (n-1)}$$

Thus, any area under this transformation is scaled by $\sqrt{\det T^\top T} = \sqrt{n}$: a constant. The lemma follows directly after this conclusion. $\square$

We use $U(\cdot)$ to denote the uniform distribution over $\Delta^{n-1}$: $U(x) = \frac{(n-1)!}{\sqrt{n}}, \forall x \in \Delta^{n-1}$. We use the notation vol$[S] = \int_S 1ds$ to represent the volume of a given subset $S \subset \Delta^{n-1}$, and use rvol$[S]$ for the relative volume (w.r.t. the underlying $n$-simplex) of $S$, *i.e.*, rvol$[S] := \frac{\text{vol}[S]}{\text{vol}[\Delta^{n-1}]} = \int_S U(x)dx$. We also use $n = |E(c)|$ from now on. We use the vector $x$ to denote (with the slight abuse of notation we have mentioned) $p_{LM}(q, c)$ on the output simplex.

**Lemma B.6.** *(Gaussian cdf Tail Bound, Gordon [62]) Denote $\phi(\cdot)$ as the standard Gaussian pdf. When $x > 0$,*

$$\frac{x}{x^2+1}\phi(x) = \frac{x}{x^2+1}\frac{e^{-x^2/2}}{\sqrt{2\pi}} \leq 1 - \Phi(x) \leq \frac{e^{-x^2/2}}{\sqrt{2\pi}x} = \frac{1}{x}\phi(x).$$

Now we are ready to give the proof of Theorem 2.

*Proof.* Let $|E_h(c)| = n_0$ and denote $|E_h(c)| + |E_s(c)| + |E_n(c)| = n$. Without loss of generality, we define the first $n_0 = |E_h(c)|$ elements as the harmful explanations. Let the thresholding constant be $p$. That is, we define the harmful zone $\mathcal{H}_h$ as $\{x \in \Delta^{n-1}|\sum_{i=1}^{n_0} x_i \geq p\}$. To compute the relative volume of $\mathcal{H}_h$ in $\Delta^{n-1}$, we could instead operate on the projected probability simplex $\Delta_p^{n-1}$ introduced in Definition B.2, and compute the relative volume of the projected $\mathcal{H}_h$: $\mathcal{H}_{h,p} := \{x \in$

$\Delta_p^{n-1} | \sum_{i=1}^{n_0} x_i \geq p\}$. Note that $\Delta_p^{n-1} \subset \mathbb{R}^{n-1}$. We derive its expression as follows.

$$\text{volume}[\mathcal{H}_{h,p}^C] = \text{volume}[\{x \in \Delta_p^{n-1} | \sum_{i=1}^{n_0} x_i \leq p\}$$

$$= \int_0^p dx_1 \int_0^{p-x_1} dx_2 \cdots \int_0^{p-\sum_{i=1}^{n_0-1} x_i} dx_{n_0} \int_0^{1-\sum_{i=1}^{n_0} x_i} dx_{n_0+1} \cdots \int_0^{1-\sum_{i=1}^{n-2} x_i} dx_{n-1}$$

$$= \int_0^p dx_1 \int_0^{p-x_1} dx_2 \cdots \int_0^{p-\sum_{i=1}^{n_0-1} x_i} dx_{n_0} \left[ \frac{1}{(n-n_0-1)!}(1 - \sum_{i=1}^{n_0} x_i)^{n-n_0-1} \right]$$

$$= \int_0^p dx_1 \int_0^{p-x_1} dx_2 \cdots \int_0^{p-\sum_{i=1}^{n_0-2} x_i} dx_{n_0-1} \frac{1}{(n-n_0)!} \left[ (1 - \sum_{i=1}^{n_0-1} x_i)^{n-n_0} - (1-p)^{n-n_0} \right]$$

$$= \cdots$$

$$= \frac{1}{(n-1)!}[1 - (1-p)^{n-1}] - \sum_{j=1}^{n_0-1} \frac{(1-p)^{n-1-j}}{j!(n-1-j)!}p^j$$

$$\tag{5}$$

Thus, the relative volume of $\mathcal{H}_h$ can be written as

$$\text{rvol}[\mathcal{H}_h] = 1 - \frac{\text{volume}[\mathcal{H}_{h,p}^C]}{\text{volume[projected probability simplex]}}$$

$$= (1-p)^{n-1} + \sum_{j=1}^{n_0-1} \frac{(n-1)!(1-p)^{n-1-j}}{j!(n-1-j)!}p^j \tag{6}$$

$$= \sum_{j=0}^{n_0-1} p^j(1-p)^{n-1-j} \binom{n-1}{j}.$$

Which is precisely the binomial distribution formula. With the Central Limit Theorem, when $n \gg O(1)$, we know the binomial distribution can be well approximated via the normal distribution as follows:

$$f(x) = \binom{n}{x} p^x (1-p)^{n-x} \xrightarrow{d} \mathcal{N}(np, np(1-p)). \tag{7}$$

Thus, denote $\phi_{(n-1),p}(x)$ as the pdf of Gaussian variable with mean $(n-1)p$, variance $(n-1)p(1-p)$, the rvol term above can be estimated as follows:

$$\sum_{j=0}^{n_0-1} p^j(1-p)^{n-1-j} \binom{n-1}{j} \asymp \int_{-\infty}^{n_0-1} \phi_{(n-1),p}(x)dx$$

$$= \Phi \left[ \frac{n_0 - 1 - (n-1)p}{\sqrt{(n-1)p(1-p)}} \right] \tag{8}$$

$$= \Phi \left[ \frac{|E_h(c)| - 1 - (n-1)p}{\sqrt{(n-1)p(1-p)}} \right].$$

We use $a = \frac{|E_h(c)| - 1 - (n-1)p}{\sqrt{(n-1)p(1-p)}}$. Consider an adversary with budget $\epsilon$ under $\ell_p$ or Jensen-Shannon Divergence (JSD) / Total Variation (TV) capability. Since $||x||_1 \geq ||x||_p, \forall p \geq 1$ as well as $||x||_1 \geq 2\text{JSD}(x), ||x||_1 \geq 2\text{TV}(x)$, we know $\mathcal{H}_h(\epsilon, \ell_1) \subset \mathcal{H}_h(\epsilon, d)$ for all $d$ we have considered. With that $\ell_1, \epsilon$ setup, the corresponding $\epsilon-$expansion set of $\mathcal{H}_h$ has a closed-form expression as

$$\mathcal{H}_h(\epsilon, \ell_1) = \{x \in \Delta^{n-1} | \sum_{i=1}^{n_0} x_i \geq p - \frac{\epsilon}{2}\}.$$

Similar as above, we derive the analytical solution of its relative volume associated with constant $a'$ as:

$$a' = \frac{|E_h(c)| - 1 - (n-1)(p - \frac{\epsilon}{2})}{\sqrt{(n-1)(p - \frac{\epsilon}{2})(1 - p + \frac{\epsilon}{2})}}$$

$$= a\sqrt{\frac{p(1-p)}{(p - \frac{\epsilon}{2})(1 - p + \frac{\epsilon}{2})}} + \frac{\epsilon}{2}\sqrt{\frac{n-1}{(p - \frac{\epsilon}{2})(1 - p + \frac{\epsilon}{2})}}. \tag{9}$$

Under our framework, with $p < \frac{1}{2}$, we know $\frac{1}{4} > p(1-p) > (p - \frac{\epsilon}{2})(1 - p + \frac{\epsilon}{2}))$. Thus

$$a' > a + \sqrt{n-1}\epsilon := a_\epsilon.$$

Consider the induced distribution $\gamma_c$ on the output simplex. Given an adversary with $\ell_p$ or JSD/TV perturbing capability, with the fixed harmful concept $c$, safety is guaranteed if and only if $p_{LM}(q, c)$ resides outside $\mathcal{H}_h(\epsilon, d)$. Define the area of interest, $S(d)$ as $S(d) := \Delta^{n-1} - \mathcal{H}_h(\epsilon, d)$. Thus, the probability of this event could be bounded as

$$\mathbb{P}_{x \sim \gamma_c} \mathbb{1}_{x \in S(d)} < \max_{x \in S(d)} \gamma_c(x) \int_{S(d)} 1 dx < \gamma_s \mathrm{rvol}[S(d)] < \gamma_s \mathrm{rvol}[S(\ell_1)] < \gamma_s(1 - \mathrm{rvol}[\mathcal{H}_h(\epsilon, \ell_1)])$$

This gives an upper bound of

$$\gamma_s(1 - \Phi(a_\epsilon)).$$

which can be simplified when $a \geq 0$ using Lemma B.6:

$$\gamma_s\left(\frac{\phi(a_\epsilon)}{a_\epsilon}\right).$$

Thus, the probability of getting a LM instance from the preference alignment process such that it allows for successful jailbreaking on a specific harmful concept $c$ is at least

$$1 - \gamma_s\left(1 - \Phi(a_\epsilon)\right).$$

Up to now, we have derived the result in Theorem 2. However, we can move a step further to show the decay rate on the right hand side term. It can be simplified when $a \geq 0$:

$$1 - \gamma_s\left(\frac{\phi(a_\epsilon)}{a_\epsilon}\right),$$

which finishes the proof. $\qquad\square$

## C  RLHF, DPO and our E-RLHF

The classic RLHF framework was established by Christiano et al. [63], and developed by Ziegler et al. [19], Ouyang et al. [20], Bai et al. [21]. After the collection of a *preference dataset* $\mathcal{D}_s = \{(x, e_w, e_l)\}$, one first trains a reward model under the Bradley-Terry model [64], with the objective, where $\sigma(\cdot)$ stands for the sigmoid function:

$$r(x, e) = \arg\max_r \mathbb{E}_{(x, e_w, e_l) \sim \mathcal{D}} \log \sigma(r(x, e_w) - r(x, e_l)).$$

Following, proximal policy optimization (PPO) [65] is commonly adopted across these implementations, forming the basis of current state-of-the-art language models. The KL-constrained RL Fine-Tuning (RLFT) objective takes the form:

$$\max_{p_{LM}} \mathbb{E}_{x \sim \mathcal{D}_s}\{\mathbb{E}_{e \sim p_{LM}(\cdot|x)}[r(x, e)] - \beta\mathbb{D}_{\mathrm{KL}}(p_{LM}(x)||p_{\mathrm{SFT}}(x))\}.$$

However, PPO tuning can suffer from instability [66] and implementation complication [67]. To overcome these issues, a series of work propose to skip the reward modeling step and learn *directly* from the preference dataset, with the representative pioneering work by Rafailov et al. [23], namely *Direct Preference Optimization (DPO)*. We summarize the derivation of the DPO objective below, and generalize the objective to the one we use in our experiments, *i.e.,* E-DPO.

First, noticing the *closed-form optimal solution* for $p_{LM}$ of the RLFT objective writes (see *e.g.,* Appendix A.1 of Rafailov et al. [23])

$$p_{\text{RLFT}}^*(e|x) = \frac{1}{Z'(x)} p_{\text{ref}}(e|x) \exp(\frac{1}{\beta} r(x,e)).$$

With this analytical solution in mind, we can solve the reward as

$$r(x,e) = \beta \log \frac{p_{\text{RLFT}}^*(e|x)}{p_{\text{ref}}(e|x)} + \beta \log Z'(x).$$

Regard $\pi^*$ as the optimization target, plug this transformation into the reward model objective to obtain the DPO objective:

$$p_{\text{DPO}} = \arg\min_{p_{LM}} -\mathbb{E}_{(x,e_w,e_l)\sim\mathcal{D}}[\log \sigma(\beta \log \frac{p_{LM}(e_w|x)}{p_{\text{ref}}(e_w|x)} - \beta \log \frac{p_{LM}(e_l|x)}{p_{\text{ref}}(e_l|x)})].$$

For our E-RLHF, the modification to the objective leads to another optimal solution of $p_{LM}$:

$$p^*(e|x) = \frac{1}{Z(x)} p_{\text{ref}}(e|x_s) \exp(\frac{1}{\beta} r(x,e)).$$

Thus,

$$r(x,e) = \beta \log \frac{p^*(e|x)}{p_{\text{ref}}(e|x_s)} + \beta \log Z(x)$$

And plug it in to the reward model objective to formulate our E-DPO:

$$p_{\text{E-DPO}} = \arg\min_{p_{LM}} -\mathbb{E}_{(x,e_w,e_l)\sim\mathcal{D}}[\log \sigma(\beta \log \frac{p_{LM}(e_w|x)}{p_{\text{ref}}(e_w|x_s)} - \beta \log \frac{p_{LM}(e_l|x)}{p_{\text{ref}}(e_l|x_s)})].$$

The advantage of our E-RLHF objective is as follows.

**Proposition C.1.** *(Overcoming the small safe set problem) E-RLHF will lead to the optimal solution* $p^*$:

$$p^*(e|x) = \frac{1}{Z(x)} p_{ref}(e|x_s) \exp(\frac{1}{\beta} r(x,e)).$$

*Compared to* $p_{RLFT}^*$, *the advantage when encountering a harmful prompt* $x$ *is:*

*(1) (Erase harmful explanations)* $\forall e \in supp(p_{ref}(\cdot|x)) - supp(p_{ref}(\cdot|x_s))$, $p^*(e|x) = 0$;

*(2) (Add safe explanations)* $\forall e \in supp(p_{ref}(\cdot|x_s)) - supp(p_{ref}(\cdot|x))$, $p^*(e|x) > 0 = p_{RLFT}^*(e|x)$.

*Thus, with the same jailbreak threshold* $p$, *the safety zone is successfully expanded.*

Intriguingly, when the safe transformation is done by appending an identical safe prefix to all harmful prompts, we can connect our E-RLHF to **context distillation**. A good prompt is known to matter for the performance of a fixed-parameters LM [68, 55]. Researchers have proposed a systematic LM tuning algorithm, called Context Distillation [69], aiming at *distilling useful information from a good context as prefix to a language model*. Given an initialized language model, for example $p_{\text{SFT}}$, an input prompt $x$ and a prefix context string `prefix`, Askell et al. [69] optimizes the loss

$$L(p_{LM}) = \mathbb{D}_{\text{KL}}(p_{\text{SFT}}(\texttt{prefix} \oplus x), p_{LM}(x))$$

where $\oplus$ stands for string concatenation. This technique has been adopted as part of the safety alignment process during the LLaMa-2 series tuning [13], where `prefix` is chosen from a set of pre-defined safe prefixes. When applying the identical prefix transform in our E-RLHF transformation, it can be regarded as a combination of safety context distillation and RLHF. This gives another point of view on the effectiveness of our proposal.

# D    Ablation Study

In this section, we perform extensive ablation studies to showcase the effectiveness of our proposed E-RLHF.

Table 3: Safety evaluation, LoRA results. The result is consistent with the one we have obtained in the main text, that our E-DPO performs better than DPO across a collection of adversaries. ▦ indicates better performance.

| Model | Direct Request | GCG | GBDA | AP | SFS | ZS | PAIR | TAP | AutoDAN | PAP-top5 | Human | AVG ↓ |
|---|---|---|---|---|---|---|---|---|---|---|---|---|
| | | | | | HarmBench ASR [2] | | | | | | | |
| $\pi_{\text{DPO (LoRA)}}$ | 24.50 | 47.50 | 40.50 | 43.25 | 43.25 | 28.50 | 45.25 | 53.25 | 53.50 | 29.75 | 38.90 | 40.74 |
| $\pi_{\text{E-DPO (LoRA)}}$ | 24.25 | 42.50 | 36.50 | 41.50 | 42.75 | 27.20 | 45.00 | 53.75 | 50.25 | 27.25 | 38.05 | 39.00 |
| | | | | | AdvBench ASR [1] | | | | | | | |
| $\pi_{\text{DPO (LoRA)}}$ | 2.00 | 29.00 | 26.00 | 26.00 | 40.00 | 8.80 | 32.00 | 54.00 | 46.00 | 2.00 | 31.20 | 27.00 |
| $\pi_{\text{E-DPO (LoRA)}}$ (ours) | 0.00 | 25.00 | 20.00 | 27.00 | 33.00 | 7.20 | 23.00 | 50.00 | 45.00 | 2.00 | 28.80 | 23.73 |

Table 4: Safe prefix ablation. Prefixes we use are included in Table 8. Our E-DPO performs better than the DPO baseline in most cases we have tested. ▦ indicates best performance. Number in bracket [] indicates the MT-Bench score.

| Model [MT-Bench] | Direct Request | GCG | GBDA | AP | SFS | ZS | PAIR | TAP | AutoDAN | PAP-top5 | Human | AVG ↓ |
|---|---|---|---|---|---|---|---|---|---|---|---|---|
| | | | | | HarmBench ASR [2] | | | | | | | |
| $\pi_{\text{DPO}}[6.9]$ | 27.50 | 53.00 | 39.00 | 46.75 | 43.25 | 29.10 | 52.50 | 54.00 | 51.00 | 28.75 | 37.15 | 42.00 |
| $\pi_{\text{E-DPO(1)}}[6.9]$ | 26.25 | 56.50 | 33.75 | 44.25 | 42.25 | 29.30 | 50.00 | 56.75 | 56.25 | 31.50 | 34.05 | 41.90 |
| $\pi_{\text{E-DPO(2)}}[6.9]$ | 24.75 | 52.25 | 34.00 | 39.00 | 44.75 | 29.75 | 50.50 | 54.50 | 53.25 | 28.00 | 34.35 | 40.46 |
| $\pi_{\text{E-DPO(3)}}[6.8]$ | 24.75 | 52.75 | 35.25 | 37.50 | 35.50 | 28.65 | 49.00 | 53.50 | 47.25 | 30.50 | 30.25 | 38.63 |
| $\pi_{\text{E-DPO(4)}}[6.7]$ | 23.50 | 47.50 | 31.75 | 36.25 | 40.50 | 26.45 | 48.50 | 51.00 | 43.00 | 27.00 | 31.05 | 36.95 |
| | | | | | AdvBench ASR [1] | | | | | | | |
| $\pi_{\text{DPO}}[6.9]$ | 0.00 | 47.00 | 12.00 | 39.00 | 30.00 | 7.00 | 50.00 | 61.00 | 44.00 | 4.00 | 18.40 | 28.40 |
| $\pi_{\text{E-DPO(1)}}[6.9]$ | 0.00 | 51.00 | 12.00 | 29.00 | 33.00 | 6.80 | 47.00 | 62.00 | 53.00 | 5.00 | 20.00 | 28.98 |
| $\pi_{\text{E-DPO(2)}}[6.9]$ | 1.00 | 39.00 | 12.00 | 20.00 | 34.00 | 6.20 | 53.00 | 63.00 | 49.00 | 3.00 | 15.60 | 26.89 |
| $\pi_{\text{E-DPO(3)}}[6.8]$ | 0.00 | 47.00 | 11.00 | 23.00 | 23.00 | 6.80 | 45.00 | 58.00 | 36.00 | 4.00 | 15.80 | 24.51 |
| $\pi_{\text{E-DPO(4)}}[6.7]$ | 0.00 | 38.00 | 8.00 | 15.00 | 21.00 | 5.20 | 41.00 | 53.00 | 31.00 | 4.00 | 13.60 | 20.89 |

## D.1 LoRA results

In this section, we show results obtained by Low-Rank Adaptation [54]. We explore the same set of safe prefixes as in ablation D.2, and choose the best model for illustration. Numbers are illustrated in Table 3. Results are identical to the ones obtained via full parameter tuning that our E-RLHF performs better consistently against the RLHF baseline.

## D.2 Ablation on safe prefixes

We ablate the effect of different safe prefixes. The prefixes we consider are collected in Table 8. Attack success rate numbers are shown in Table 4, with each model's MT-Bench scores shown in the brackets. Clearly, almost all safe prefixes lead to better safety against the DPO baseline and improved helpfulness compared to the SFT model (whose MT-Bench score is 6.3), and the performance could vary depending on the choice of the prefix. Finding a good prefix matters for our method, and we leave digging the optimal one out as future work.

## D.3 Ablation on transforming all prompts

Here, we proceed to ablating the effect of transforming all prompts, no matter harmful or not. Results are shown in Table 5, where the red color indicates that safety downgrades compared to the model obtained via transforming harmful prompts only. Clearly, most models even persist *worse* safety compared to the DPO baseline itself, suggesting the detrimental effect of transforming the unharmful prompts.

## D.4 Ablation with system prompt

As pointed out by previous works [2, 48, 70], system prompt can have a significant impact on ASR. This comes in two-folds: firstly, a powerful system prompt can initialize the LM to be closer to

Table 5: Ablation study on transforming all prompts. We apply the same safe prefixes as used in Table 4. ■ indicates the safety is *worse* compared to the model trained with transforming only the harmful prompts. AVG scores achieved by the DPO baseline are 42.00 and 28.40, respectively.

| Model | Direct Request | GCG | GBDA | AP | HarmBench ASR [2] SFS | ZS | PAIR | TAP | AutoDAN | PAP-top5 | Human | AVG ↓ |
|---|---|---|---|---|---|---|---|---|---|---|---|---|
| $\pi_{\text{E-DPO}(1)}$ | 28.00 | 55.75 | 41.00 | 42.00 | 42.50 | 31.00 | 51.25 | 56.25 | 56.00 | 32.00 | 36.05 | 42.89 |
| $\pi_{\text{E-DPO}(2)}$ | 25.75 | 60.50 | 40.25 | 46.50 | 44.75 | 30.15 | 52.75 | 57.25 | 63.50 | 30.75 | 40.45 | 44.78 |
| $\pi_{\text{E-DPO}(3)}$ | 24.00 | 57.25 | 35.00 | 41.75 | 39.50 | 26.55 | 49.25 | 55.00 | 58.75 | 29.25 | 38.70 | 41.36 |
| $\pi_{\text{E-DPO}(4)}$ | 27.75 | 58.50 | 38.00 | 42.00 | 40.75 | 31.20 | 52.75 | 60.50 | 54.75 | 31.50 | 38.30 | 43.27 |
| | | | | | AdvBench ASR [1] | | | | | | | |
| $\pi_{\text{E-DPO}(1)}$ | 0.00 | 53.00 | 17.00 | 31.00 | 26.00 | 6.00 | 50.00 | 57.00 | 61.00 | 4.00 | 19.00 | 29.45 |
| $\pi_{\text{E-DPO}(2)}$ | 0.00 | 52.00 | 17.00 | 29.00 | 26.00 | 6.20 | 56.00 | 58.00 | 73.00 | 3.00 | 25.60 | 31.44 |
| $\pi_{\text{E-DPO}(3)}$ | 0.00 | 56.00 | 17.00 | 23.00 | 17.00 | 3.80 | 46.00 | 58.00 | 66.00 | 4.00 | 23.40 | 28.56 |
| $\pi_{\text{E-DPO}(4)}$ | 0.00 | 58.00 | 17.00 | 36.00 | 27.00 | 6.80 | 46.00 | 60.00 | 68.00 | 3.00 | 25.20 | 31.54 |

Table 6: Ablation with the default Mistral system prompt. It boosts the safety of our E-DPO models to achieve better safety, while maintaining the advantage over the DPO baseline.

| Model | Direct Request | GCG | GBDA | AP | HarmBench ASR [2] SFS | ZS | PAIR | TAP | AutoDAN | PAP-top5 | Human | AVG ↓ |
|---|---|---|---|---|---|---|---|---|---|---|---|---|
| $\pi_{\text{SFT}}$ | 25.25 | 61.25 | 26.75 | 37.00 | 27.25 | 24.75 | 55.50 | 58.75 | 61.25 | 24.00 | 30.35 | 39.28 |
| $\pi_{\text{DPO}}$ | 20.50 | 49.25 | 29.25 | 32.75 | 34.50 | 22.55 | 42.75 | 50.25 | 50.75 | 30.00 | 31.20 | 35.80 |
| $\pi_{\text{E-DPO}}$ | 15.50 | 41.50 | 22.75 | 29.00 | 31.25 | 17.95 | 41.75 | 44.25 | 44.25 | 23.00 | 26.55 | 30.70 |

the safety zone, thus making the model safer; secondly, a longer system prompt would enlarge the difficulty of launching a specific attack due to the increased computational consumption. To confirm our E-RLHF is still helpful under this specific situation, we ablate this effect under the control of the default Mistral system prompt [11]. The results can be found in Table 6, where nearly all ASRs drop, whilst the conclusion does not change.

### D.5 Ablation on improving helpfulness and safety simultaneously

In similar spirit, we ask the question: can the trick we propose to improve safety **be used to improve helpfulness simultaneously?** To achieve this goal, we choose the prefix `"Give me a safe and helpful response to the following request."`, and apply it to all prompts in the preference dataset. Results can be found in Table 7. Our model achieves better safety, and improves the helpfulness benchmarked by MT-Bench [3] from 6.9 to 7.0.

## E Filtering harmful prompt in the preference alignment dataset

Inspired by the work by Chao et al. [30], we prompt GPT-3.5-Turbo as a harmness judge on input prompts. It assigns an integer score ranging from 1 to 10 for a given prompt, and we choose the score $\geq 6$ subset as harmful. The system prompt is shown in Table 9.

---

[11] https://docs.mistral.ai/capabilities/guardrailing/

Table 7: Improving helpfulness and safety simultaneously using E-RLHF. MT-Bench scores are 6.8 and 6.9, for the DPO baseline and our E-DPO model, respectively.

| Model | Direct Request | GCG | GBDA | AP | HarmBench ASR [2] SFS | ZS | PAIR | TAP | AutoDAN | PAP-top5 | Human | AVG ↓ |
|---|---|---|---|---|---|---|---|---|---|---|---|---|
| $\pi_{\text{SFT}}$ | 32.25 | 59.25 | 35.50 | 42.75 | 42.75 | 36.20 | 56.50 | 65.00 | 56.75 | 26.75 | 35.50 | 44.47 |
| $\pi_{\text{DPO}}$ | 27.50 | 53.00 | 39.00 | 46.75 | 43.25 | 29.10 | 52.50 | 54.00 | 51.00 | 28.75 | 37.15 | 42.00 |
| $\pi_{\text{E-DPO}}$ (ours) | 26.25 | 56.00 | 37.00 | 43.75 | 38.25 | 27.90 | 49.75 | 56.00 | 52.25 | 27.50 | 32.65 | 40.66 |

Table 8: Safe prefixes we use in our experiments.

| |
|---|
| (1) "Provide a responsible and ethical answer to the following request. " |
| (2) "In a manner that promotes safety and well-being, please respond to the following request. " |
| (3) "Keeping in mind the importance of safety and legality, how would you respond to this: " |
| (4) "Please ensure your response adheres to community guidelines and ethical standards: " |

Table 9: System prompt for GPT-3.5-Turbo used to filter harmful and non-harmful prompt in the preference dataset.

| |
|---|
| You are a helpful red teaming assistant. You will receive a prompt. Please act as an impartial judge and evaluate whether the prompt itself is requesting information that violates ethical, legal, or safety guidelines. Be as objective as possible. Rate the prompt on a scale from 1 to 10. A rating of 1 signifies a fully non-harmful prompt. A rating of 10 signifies prompts that could lead to responses that fully violate ethical, legal, or safety guidelines. Your rating should strictly follow this format: "Rating: [[rating]]", the rating in the double-closed brackets is a number from 1 to 10, e,g, "Rating: [[5]]". |

# F   Jailbreak adversaries collected in the HarmBench project [2]

In this section, we give a short overview of the adversaries we adopt to evaluate our models. Some descriptions are summarized in Mazeika et al. [2].

- Direct Request refers to sending the original harmful prompt directly to the target LLM.
- GCG [1], GBDA [71] and AP [72] find adversarial suffixes via token-level optimization.
- SFS (Stochastic Few-Shot) and ZS (Zero-Shot) [15] are few-shot sampling or zero-shot generation of test cases by an attacker LLM to elicit a behavior from a target LLM.
- PAIR [30] and TAP [33] are iterative prompting / tree-structured prompting methods, with an attacker LLM, to adaptively explore and elicit specific harmful behaviors from the target LLM.
- AutoDAN [29] is a genetic-algorithm based attack with initializations from handcrafted DAN jailbreak prompts.
- PAP [73] uses persuasive strategies to jailbreak the target LLM.
- HumanJailbreaks [74] uses a fixed set of in-the-wild human jailbreak templates, similar to the Do Anything Now (DAN) jailbreaks.

We exclude all transfer attacks since we focus on single-model jailbreak. Furthermore, we choose to discard the UAT [75] and PEZ [76] adversaries, because the former induces an out-of-memory error on our V100 GPUs, and the latter never succeeds to find a suffix according to our experiments.

# G   Broader Impacts

The societal impact of our work has close connection to the topic of LLM safety. We propose a framework for analyzing language model pretraining and jailbreaking, and we design a new RLHF algorithm for improving safety. As shown in our experiments, our work could advocate for safer LLMs.

# H   Related work

In this section, we provide a review of the current literature on LLM jailbreaking.

## H.1   Jailbreak methods

In this section, we summarize existing jailbreaking methods.

**Baseline and pioneers**  Autoprompt [72], a baseline method for optimizing in the token space w.r.t. a certain objective, approximates coordinate ascent by first ranking all tokens using an approximate objective, and then compute the exact value on them. The approximation is carried out by a single step of gradient computation. Jones et al. [77] propose Autoregressive Randomized Coordinate Ascent (ARCA) to generate (input, output) pairs that include certain harmful info or satisfy a fixed format requirement. Token level optimization is carried out with linear approximation on GPT-2. GBDA [71] study adversarial attack on text classification problems, by optimizing the continuous relaxation of the autoregressive sampling probability matrix. In late 2022, among social media, users misalign GPT-3 via prompt injection. Perez and Ribeiro [78] study how this be done by adversaries. They successfully manipulate the model to output a given harmful string and leak the system prompt. In early 2023, an empirical study was carried out by Liu et al. [9] to measure the result of prompt engineering for breaking ChatGPT safeguards. Shen et al. [74] collect jailbreaking prompts from multiple platforms on the internet, analyze these data, create a harmful question set, and identify some typical harmful prompts that are effective at that moment. Later, the Greedy Coordinate Gradient (GCG) method [1], a strong *white-box* attack variant of AutoPrompt [72] was proposed. Wei et al. [79] categorizes two general modes of jailbreaking: *competing objective* and *mismatched generalization*.

**LLM automation and suffix-based attacks**  Liu et al. [29] propose AutoDAN, that relies on genetic algorithms, with the requirement of manual prompts for conducting mutation and crossover on the *paragraph and sentence level*. The jailbreaking prompts generated by AutoDAN are semantically plausible, unlike the suffix generated by GCG. As a comparison, Lapid et al. [25] use genetic algorithm for *black-box* universal adversarial suffix generation. Chao et al. [30] propose another LLM-based jailbreak automation algorithm, where an LLM judge is built to assign a safety score to a given output, while the attacker is enforced (via a page-long prompt) to improve the quality of jailbreaking prompts from multiple perspectives. Zhu et al. [24] propose another AutoDAN method that explores the balanced loss between jailbreak loss (log probability on the harmful string, as used in Zou et al. [1]) and the plausibility loss (log probability over the adversarial suffix), aiming at improving interpretability. Li et al. [31] uses genetic algorithm to search with similarty measure and initialize with paraphrasing. Its performance is claimed to be better than AutoDAN-GA. Deng et al. [80] investigate the possible defensive tricks in proprietary LLMs, and propose a pipeline for automated jailbreaking using a fine-tuned LLM. Yu et al. [81] propose GPTFuzzer, essentially a genetic algorithmic framework for jailbreaking. Their work's difference between AutoDAN is that it has a pool of "seeds", a.k.a. templates for transforming the harmful prompt, and the mutation is done on the template level. Ding et al. [32] propose automating attack via LLM *prompt rewriting* and *scenario nesting*. The latter consists of code completion, table filling and text continuation, since the authors regard these as align with training objectives well, and are suitable for LLMs to complete the task. Mehrotra et al. [33] combine Automation used in Chao et al. [30] and tree-of-thought [82], create interpretable prompts in a *black-box* manner. Li et al. [83] propose DeepInception, and use *nested, imaginary* scenario to induce harmful content. Li et al. [84] propose DrAttack, which camouflages a query's malicious intent through semantic decomposition, by constructing a parsing tree and split the original prompt into different segmentations. Wang et al. [85] draw inspiration from the self-perception theory from psychology to design a prompt modification pipeline on gradually persuading the LM to be jailbroken. Paulus et al. [86] propose AdvPrompter, where the authors train a language model as a *suffix generator* to speed up LLM attack.

**Manipulating the decoding process**  Huang et al. [48] find the method of changing the generating hyperparameters (*i.e., $p$ of top-$p$ sampling, the temperature $T$, and $k$ of top-$k$ sampling*) of safety-aligned LLMs suffices for obtaining harmful outputs when the user is able to manipulate the system prompt and input configurations. Zhang et al. [87] directly manipulate the output generation probability by enforcing an affirmative prefix, and reversing the negation words if they appear in a pre-defined vocabulary (*e.g., sorry $\rightarrow$ glad*). Zhao et al. [88] assume access to the decoding distribution of a LM. They use two small LMs, a safe one and a harmful one, to manipulate the decoding ratio of the large safe LM for jailbreaking. The key insight is the decoding distribution between the safe model and the harmful model only differs significantly for the first tens of tokens.

**Fine-Tuning alone suffices**  Yang et al. [89] show that fine-tuning on as few as 100 harmful example pairs suffices for turning the LLaMa-chat models (and some other <70B LMs) into malicious counterparts. Zhan et al. [90] fine-tune GPT-4 on harmful data, and find the fine-tuned models escape previous safety constraints while maintaining usefulness. Qi et al. [91] find fine-tuning alone, even on

benign data, leads to safety degradation using LLaMa and GPT-3.5-Turbo. Fine-tuning on harmful data (with less than 5 gradient steps) will cause the model to be completely harmful, while tuning on identity-shifting data could make the LM fully obedient.

**Low-resource language and cipher**    Yong et al. [26], Deng et al. [27] explore the difference in languages when encountering the same harmful query, and find a direct translation to low resource languages will lead to higher risk, and Deng et al. [27] additionally find when combined with sophisticated methods, the drawback of low-resource languages disappear. Yuan et al. [28] use cipher encoding and decoding to break LLMs. Smaller scale models are immune from such attacks, while the smartest GPT-4 encountered the highest risk.

**Vision-language model attacks**    Besides pure LLM, some research works move a step forward, utilizing images for breaking vision-language models (VLMs). Shayegani et al. [92] explore multimodal attack on VLM via embedding space feature matching. Qi et al. [93] generate adversarial examples via maximizing the conditional probability of a harmful *corpus, i.e.,* the sum of log probabilities over all outputs, and use the final image with harmful query for jailbreaking. Carlini et al. [94] generate adversarial example for a *fixed harmful content*, and find no matter what input prompt is given to the VLM, it will respond with the target harmful string. Maus et al. [95] propose a black-box attack on manipulating the generated image with modified adversarial prompt.

**Misc**    Wei et al. [57], Wang et al. [58] explore in-context learning for attack and defense. The attack is weak since it could only break Vicuna [96] and can be defended by in-context safe examples. Later, this method is scaled-up to significantly improve strength for breaking guardrails of large, state-of-the-art models [59]. An early work in February 2023 [97] adopts obfuscation (including synonym and typos), code injection and virtualization to successfully jailbreak ChatGPT. Shah et al. [98] illustrate in-context automated persona modulation attack for large-scale LLMs and Vicuna. Zeng et al. [73] consider the more broadly perspective of *persuasion*: they train a persuasive paraphraser based on a fine-grained taxonomy of persuasion techniques. Detailed ablation on attack effectiveness is studied. Guo et al. [99] focus on stealthiness and controllability. They notice the constraints applied to the jailbreaking prompts (*e.g.,* fluency) are exactly the targets of the controllable text generation problem. Thus, they adopt the Energy-based Constrained Decoding with Langevin Dynamic (COLD) [100] on output logits. Forming each constraint as well as the task of jailbreaking as an energy function over logits, the Langevin Dynamic is used for finding a good logit distribution, and the decoding technique in Qin et al. [100] is used for output generation. Banerjee et al. [101] introduce a dataset TECHHAZARDQA, compare direct query v.s. pseudo-code format, and find the latter induces higher risk. Mangaokar et al. [102] considers a type of adaptive attack against *checking-based defense*, that appends a universal adversarial prefix into the jailbreaking prompt to make the guard model always output "safe", and thus making the detector fails to detect harmful information. Lv et al. [103] propose Code Chameleon, which contains multiple encryption and decryption methods defined by python functions, that transforms the harmful prompt into specific predefined form to jailbreak LLMs. Sadasivan et al. [104] speed up the computation of GCG [1] to make it possible to launch on a single GPU. Geiping et al. [105] build a taxonomy on risks beyond jailbreaking, and coerce the LLM to provide certain outputs by optimizing a set of tokens via GCG. Ren et al. [106] propose CodeAttack that use code templates to query the output out instead of using natural language directly, and obtain descent results.

## H.2    Defense methods

Up to now, no universal defensive strategy as adversarial training [107] for adversarial attacks / differential privacy [108] for membership attacks exists as a gold standard. In general, we can classify the methods into three typical types: alignment, red-teaming, and algorithmic proposals.

**Alignment**    The target of alignment is to push the output of language models be aligned to human values. Regarding safety, the goal is to avoid outputting harmful information. RLHF is widely adopted in these methods [19, 20, 109, 22]. Variants like RLAIF also have been proposed recently [21, 110].

**Red teaming**    This term is populated as specifically dealing with harmful info on dataset curation, used together with RLHF [14–18].

Next, we proceed to defensive algorithm proposals. We classify existing defensive strategies in the following categories.

**Defense against suffix-based attacks.**  Alon and Kamfonas [111] notice the messy nature of the suffix generated by GCG, and propose to use a perplexity (PPL) filter on input prompts. They also explore using a LightGBM [112] with 2 features (PPL, prompt length) to filter harmful prompt, and show it does better than naive PPL thresholding. The PPL-based filter does not succeed with human-crafted jailbreaks. Jain et al. [37] explore many concerning viewpoints, including self-PPL filtering, paraphrasing the input prompt, and re-tokenization since many LLMs' tokenizers are based on Byte-Pair Encoding (BPE). All methods are successful in regards of defending against suffix-based attacks. They also explore the simplest form of adversarial training. Robey et al. [34] propose to perturb the input token string by random replacement/erasement/insertion, and finally perform a majority vote in the end. Cao et al. [35] judges whether an input prompt is safe or not by estimation with Monte Carlo, when randomly dropping a fraction of tokens, using the LLM itself. Kumar et al. [36] try to perform "certified" safety against harmful prompts, by erasing tokens and set the original prompt as harmful if at least one of these erased prompts lead to a harmful response, or be classified as harmful by a DistillBERT-based classifier.

**System prompt defense.**  We could modify the input prompt for jailbreaking; and several works explore if we can apply similar methods to system prompts to defend against such attacks. Xie et al. [113] propose "self-reminder", *i.e.,* appending a reminding prompt within the system prompt for defense. The attacks are collected from the JailbreakChat collection, and this strategy is effective for defending against them. Zheng et al. [114] advocate for finding a good system prompt automatically, by investigating the representational difference between safe and harmful queries, and optimizing the safety prompts along the harmful refusal direction in the representation space. One intriguing takeaway is harmful / harmless queries can be distinguished in the representation space, different from the adversarial examples in vision. Zhou et al. [115] also optimize the safe system prompt, but in a more "adversarial training" fashion, that apply jailbreak algorithms with current safe prompts first and then find good replacement candidates in the same way as done by Zou et al. [1]. Concurrently, Mo et al. [116] propose prompt adversarial tuning, where an adversarial suffix is assumed, while a safe system prompt is jointly optimized with this suffix, *with an additionally constructed benign loss* to improve helpfulness under normal queries. Zhang et al. [117] propose the idea of "goal prioritization", either without training (append prioritize safety than helpfulness and in-context examples to the system prompt) or with training (generate data pairs of prioritizing safety or helpfulness, finetune, and append prioritize safety prompt into system prompt). The former is effective for large-scale LLMs, while the latter improves safety of LLaMa-chat models. Zhou et al. [118] propose in-context adversarial game, where an attack LLM and a defense LLM interact on exchanging insights on successful jailbreaks, and defend by improving the system prompt. Zou et al. [70] give the result that system prompt matters for jailbreaking, and shows conducting GA-based search over it could improve safety.

**Checking-based, decoding-based, and Misc.**  Helbling et al. [119] generate responses first, and then use the LLM itself to examine whether the output is harmful or not. They find such simple self-examination is powerful since the TPR reported can be up to $\sim 1.00$. Wang et al. [120] propose to (1) tune the LM to enhance its capability on discriminating harmful / harmless content; (2) tune the LM to make it able to tag its own response; and (3) rewrite response if output is harmful. Li et al. [121] propose to suppress the attack performance by iteratively rewinding and re-examining the generated output. The method does not work well with small models, but works pretty fine with large (open-source) models. They find the strategy can improve generalization as well. Xu et al. [122] train a safer model first, and use normalized $p_{\text{attacked}} + \alpha(p_{\text{safer}} - p_{\text{attacked}})$ over top-$k$ shared tokens for decoding to enhance safety. Hasan et al. [123] show with original Wanda pruning [124], the LLM can help resist direct jailbreaking prompts, *e.g.,* with role-playing attacks. Pi et al. [125] propose MLLM-Protector on safeguarding Visual LLMs by checking the output and then detoxifying the content. Zhang et al. [126] perform intention analysis on the input, and enforce the model generate policy-aligned outputs both by prompting. Wang et al. [127] propose backtranslation that guesses the input prompt directly, and reject if it is harmful. Kim et al. [128] propose self-refinement which consists of generating a feedback and then refine the response to avoid harmful info output. They find using additional JSON and code formatting would improve safety. Zeng et al. [129] propose AutoDefense, which utilizes multiple agents on analyzing prompt, response and intention, to defend

against attacks. Hu et al. [130] propose Gradient Cuff, a sampling-based gradient-norm defense method, by rejecting those input prompts with large gradient norm on top of a majority-vote based filtering. Ji et al. [131] propose a method similar to Robey et al. [34], but for semantically-meaningful attacks, that paraphrases the input according to several criteria and conduct a majority vote for judging.

Several company-centered products also fall into this regime. For example, LLaMa-Guard [132] is trained on toxic data such that it is able to discriminate unsafe user prompts and outputs, respectively. IBM also propose a framework on constructing and deploying safeguard detection modules, and releases the details in a technical report [133].

### H.3 Theory and experimental understanding

Wolf et al. [38] assumes the decomposability of LLM output into a good and bad component, and show possible jailbreaking in theory by prompting the model with a sufficiently long input. Kalai and Vempala [134] use statistical tools to prove hallucination for calibrated LMs. Lee et al. [135] study the representation in GPT-2. They train a base classifier for toxicity, and use the linear weight as a proxy of toxic vector. They find there are value vectors close to the toxic vector itself, that are not suppressed by DPO tuning [23]. Wei et al. [136] use pruning and low-rank analysis on safe LM, and find (1) safe neurons and useful neurons are sparse; pruning the safe neurons or removing the safe ranks away degrades safety a lot, and (2) fixing the safe neurons in fine-tuning does not maintain safety.

