# OpenReview forum: "Mission Impossible: A Statistical Perspective on Jailbreaking LLMs"
_NeurIPS.cc/2024/Conference — NeurIPS 2024 poster_

### Official Review · Reviewer_sD4q · 2024-06-26

**Soundness:** 3
**Presentation:** 2
**Contribution:** 4
**Rating:** 7
**Confidence:** 3

**Summary:**

The paper introduces a new theoretical framework that views prompts as combinations of concepts and queries, allowing for a detailed analysis of how and why LLMs can be manipulated into producing unsafe responses. It proposes a statistical metric for alignment, focusing on quantifying the safety of the model's outputs.

The study introduces an improved alignment strategy, Enhanced Reinforcement Learning from Human Feedback (E-RLHF), which modifies existing RLHF methods to increase the likelihood of safe responses from LLMs without additional training costs.
Through experiments using standard benchmarks like AdvBench and HarmBench, the paper demonstrates that E-RLHF significantly outperforms traditional RLHF in resisting jailbreaking attempts, reducing the models' vulnerability to producing harmful outputs.

**Strengths:**

The paper offers a unique statistical framework that conceptualizes input prompts into concepts and queries. This approach allows for a nuanced understanding of how LLMs process inputs and why they might generate unsafe outputs even under alignment efforts.

The introduction of E-RLHF  as an alignment strategy that doesn't require additional computational resources is a major strength.

The empirical tests using established benchmarks such as AdvBench and HarmBench provide solid evidence that E-RLHF can significantly reduce the Attack Success Rate (ASR) compared to traditional RLHF.

**Weaknesses:**

While E-RLHF improves alignment in controlled experimental conditions, its effectiveness in more complex conversational scenarios, where inputs can evolve over a series of interactions, remains untested.

**Questions:**

The two datasets used in the work contain harmful requests about various concepts (e.g. drug, weapon). Have you tried to analyze the performance of E-RLHF on different concepts?

**Limitations:**

Yes

---

> ### Author Rebuttal · Authors · 2024-08-07
>
> Dear reviewer sD4q,
>
> Thanks for your detailed review! Here are our responses to your concerns.
>
> *1. ...(E-RLHF's) effectiveness in more complex conversational scenarios, where inputs can evolve over a series of interactions, remains untested.*
>
> - That is a great point that we had discussed internally too. We fully acknowledge that our theoretical framework does not cover multi-round chat jailbreak senarios. We have discussed this limitation in our paper (in lines 376-383, point (2)). Apart from the fact that we do currently not know how to extend the framework, we thought the exclusion of multi-interaction scenarios is justified because emirical work has not yet produced any (or SOTA) jailbreaking  attacks and evaluations:
>   - (1) **No available evalution benchmarks for multi-chat.** As discussed in the HarmBench paper [3], jailbreak evaluation varies significantly across studies, making it difficult to compare the efficacy of different methods. Therefore, we chose to follow HarmBench as our evaluation protocol, as it is designed to best reflect LLM safety by ensuring fair comparison and providing diverse jailbreak adversaries. Multi-interaction benchmarks are not available as of now.
>   - (2) **Lack of empirical methods.** The only works that use multi-round interaction to jailbreak LLMs that we are aware of are [1] and [2]. However, both were published in February and April of 2024 hence were too concurrent to integrate in our work. Additionally, both projects do not provide source code to reproduce their results, making it challenging for us to include their methods in our empirical evaluations.
>
> That being said, we believe that integrating multi-interaction attacks into our framework is crucial for future research. From a theoretical perspective, we hypothesize that if even a single attack poses a non-negligible risk, a multi-interaction attack will likely be even more challenging to defend against. Consequently, research questions would focus on how rapidly success rates escalate with the number of interactions. This exploration will provide deeper insights into the dynamics of jailbreak attack and inform more robust defensive strategies.
>
> *2. The two datasets used in the work contain harmful requests about various concepts (e.g. drug, weapon). Have you tried to analyze the performance of E-RLHF on different concepts?*
> - Our empirical evaluations primarily concentrate on harmful behavior. As highlighted in the general response, HarmBench assesses the safety of LLMs across a variety of harmful behaviors. While alignment extends beyond mere safety to include aspects such as ethical behavior, to our knowledge, we currently lack benchmarks for testing these broader criteria. We are happy to keep discussing on this topic and would welcome any specific proposals or suggestions for additional benchmarks that could enhance our understanding and assessment of ethical alignment in LLMs.
>
> We hope these explanations address your questions and concerns. Please let us know if you need further clarification, we would be delighted to discuss further.
>
> References
>
> [1] Great, Now Write an Article About That: The Crescendo Multi-Turn LLM Jailbreak Attack
>
> [2] Leveraging the Context through Multi-Round Interactions for Jailbreaking Attacks
>
> [3] HarmBench: A Standardized Evaluation Framework for Automated Red Teaming and Robust Refusal

---

> ### Comment · Reviewer_sD4q · 2024-08-11
>
> Thanks. I will keep the rating since it is already the highest.

---

> > ### Author Response · Authors · 2024-08-12
> > **Response to comments**
> >
> > Dear reviewer sD4q,
> >
> > We appreciate the recognition of our efforts and thanks for responding to our rebuttal!

---

### Official Review · Reviewer_UXGw · 2024-07-02

**Soundness:** 2
**Presentation:** 3
**Contribution:** 3
**Rating:** 6
**Confidence:** 2

**Summary:**

The paper presents a statistical framework that provides a theoretical analysis of the jailbreaking problem in language models. The authors first examine the PAC-Bayesian bound to demonstrate that there is a non-trivial probability for LLMs to mimic harmful behavior if such information is present in the pre-training corpus. They then theoretically prove that a jailbreaking prompt can be found for such pretrained LMs and propose mitigating the jailbreak issue by modifying the harmful prompts.

**Strengths:**

The paper's formulation and assumptions are clear and well-motivated. The proofs seem comprehensive and support the claims well. However, I don't have much background and PAC-Bayesian theory, so it's hard for me to verify if (1) the proofs are all rigorous (2) the proposed framework is truly novel instead of simple application of existing theorems.

The experimental results from the proposed E-RLHF improve upon DPO-based fine-tuning.

**Weaknesses:**

I find the connection between the theoretical part and section 5 (E-RLHF for actual experiment) quite handwavy. Expanding the safety zone is the claimed goal, but the proposed solution (through injecting safe concepts in a harmful prompt) seems really hacky. For example, as shown in Table 4, using safe prompt (4) gives much better results than safe prompt (1) for no reason. Overall, this proposed method seems like a small trick that the authors try to incorporate into the paper just to show some empirical value of their theoretical analysis. I also find that with such modifications, the MT-Bench score (briefly mentioned in line 355) is lower under E-DPO than DPO.

Lastly, this might be a biased opinion (that's why I didn't factor this part into my scoring decision and still gave a marginally above accept score): I feel that after all the theoretical proofs (despite their elegance), the conclusion (LLMs will mimic toxic behavior if such toxic content is present in the training corpus, jailbreak is unpreventable under reasonable assumptions, and by expanding the safe zone we reduce the probability of jailbreaking) is very intuitive and does not provide much additional insight into the problem. Maybe the mathematical formalization of the jailbreak problem itself is meaningful, and I will let the other reviewers judge the novelty of such a framework.

**Questions:**

Could you provide more details on the MT-bench's performance? For example, you have an ablation study on different safety prefixes' effects in Table 4. What about their impact on MT-bench?

In line 327 and the ablation study in the appendix, you show that non-harmful prompts should not be modified (otherwise, the model performs much worse). Could you relate this phenomenon to the safe/harmful zone or your theoretical analysis? I don't see why adding a safe prefix to a non-harmful prompt negatively impacts the safe/harmful zone, and under your assumption, this should not lead to worse behavior.

**Limitations:**

No concerns on the limitations

---

> ### Author Rebuttal · Authors · 2024-08-06
>
> Dear reviewer UXGw,
>
> Thanks for your comprehensive review! Here are our responses to your concerns.
> - **(1) The novelty of our framework.** To the best of our knowledge, we are the first to offer a theoretical analysis on jailbreaking from a statistical perspective. To tackle this problem and provide insights, we addressed the following unique challenges associated with LLMs:
>     - Abstract the capability of an LLM to generalize on unseen but plausible prompts. **This property is particularly important since many representative jailbreak methods exploit this to bypass the guardrails of existing LLMs.**
>     - Model the capability of an adversary. For jailbreak attempts, it is hard to formulate an adversary in an mathematical way, since the operations performed on prompts are discrete and unbounded.
>     - Discriminate harmful prompts versus non-harmful prompts. The statements should be connected to harmful prompts only, while the definition of "harmful" itself remains lacking.
>
> Our framework overcomes these problems and provides theoretical evidence on the difficulty and impossibility of completely avoiding jailbreaking events.
> - **(2) Connection from our theory to E-RLHF.** Under our framework, we identified a drawback in the RL fine-tuning objective, particularly the small safety zone it creates. We propose E-RLHF to address this issue by **replacing** harmful concepts in prompts $x$ with safe ones. We want to clarify that the safe prefixing strategy is **not injecting a safe concept**, but rather a **simple yet effective implementation of harmful concept replacement** in harmful prompts. We agree that more sophisticated methods could further enhance safety, and our experimental results suggest that even our simple implementation significantly improves safety. Thus, we argue our experimental results should not be regarded as negative but instead in line with Occam's razor as a positive result.
> - **(3) Sensitivity of the safe prefix and MT-Bench scores.** As mentioned in response part (1), our safe prefixing is a simple implementation of replacing harmful concepts with safe ones. Under our framework, different safe prefixes can induce different safe concepts, and different $p_{\textrm{SFT}}(x_s)$. This explains the sensitivity to the choice of safe prefix. Prompt sensitivity has also been observed in previous works ([4][5]), and we believe safety could be further improved with prompt engineering to find better safe prefixes. Regarding the MT-Bench score, we acknowledge that E-RLHF leads to a slightly lower score than RLHF. However, **the score achieved by E-RLHF is still higher than that of the SFT model**, indicating that we do not sacrifice utility for safety. There are few LLM-tuning-based defenses against jailbreak attacks, especially those improving the RLHF step. For instance, R2D2 from HarmBench [1] uses a SFT strategy, resulting in a drop in the MT-Bench score from 7.3 to 6.0. The tension between safety and utility has been noted in previous works (e.g., [2]), and our proposal does not sacrifice utility to achieve better safety.
> - **(4) Insight provided by our framework and experiments.** We aim to establish a framework that explains the jailbreak phenomenon theoretically. We argue that the intuitive results, showing that LLMs can be jailbroken both after the pretraining stage and after the current safety alignment stage with RLHF, are non-trivial, important, and counterintuitive. RLHF is the default optimization strategy for LLM alignment, and we identify a fundamental drawback in its mathematical objective. Based on this insight, we offer a plausible solution, E-RLHF, and demonstrate its effectiveness through a simple yet effective implementation.
> - **(5) Additional MT-Bench results.** We are in the process of requesting credits with the OpenAI API and plan to initiate the benchmark test as soon as possible.
> - **(6) Relating ablation study on non-harmful prompts with theoretical analysis.** Under our framework, the harmful and safety zones are defined with respect to **a single concept**, meaning modifications to non-harmful prompts should not impact performance on harmful prompts. However, we argue that the ablation phenomenon occurs due to the following reasons. Firstly, we do not model **correlations and interactions between concepts**. Each concept is considered independent, but in reality, generalization on one concept influences LLM performance on other prompts. For non-harmful prompts, appending the safe prefix may hinder optimization, affecting learning on harmful prompts. Modeling correlations and interactions between different concepts is highly complex and is left for future exploration. We will add this discussion to the limitations section of our final draft. Secondly, we point out that this statement also holds true for normal RL fine-tuning. Analysis on it (e.g., the motivation of DPO [3]) suggests that the output of a converged LLM for any prompt $x$ depends only on the reward model $r(x,e)$ and the initial distribution $p_{\textrm{SFT}}(x)$. However, due to the discrepancy between the optimal solution and the converged LLM in practice, the same reward model and LLM initialization can lead to models with significantly different performances.
>
> Lastly, we want to emphasize the significance of our experimental result. We achieve significant safety improvement across all categories of harm on 2 benchmarks. The dataset size and harm diversity coverage is the largest compared to previous papers (we refer to our general response, and discussions in [1] as reference).
>
> References
>
> [1] HarmBench: A Standardized Evaluation Framework for Automated Red Teaming and Robust Refusal
>
> [2] Training a Helpful and Harmless Assistant with Reinforcement Learning from Human Feedback
>
> [3] Direct Preference Optimization: Your Language Model is Secretly a Reward Model
>
> [4] Chain-of-Thought Prompting Elicits Reasoning in Large Language Models
>
> [5] Large Language Models as Optimizers

---

> > ### Comment · Reviewer_UXGw · 2024-08-11
> > **Official Comment by Reviewer UXGw**
> >
> > Dear Authors,
> >
> > Thanks for providing additional clarifications to the points I asked in the review. After reading the response (for both my review and others), I decide to increase the score (from 5) to 6.

---

> > > ### Author Response · Authors · 2024-08-12
> > > **Response to comments**
> > >
> > > Dear reviewer UXGw,
> > >
> > > Thanks for responding to our rebuttal and raising your score! We apologize for the delay in accessing the OpenAI API, and will include the requested comparison in our final draft.

---

### Official Review · Reviewer_T1QG · 2024-07-09

**Soundness:** 3
**Presentation:** 3
**Contribution:** 2
**Rating:** 4
**Confidence:** 2

**Summary:**

The paper addresses a very important issue of our time, the safety of LLMs. LLMs are already used in various applications and will be present in more applications to come, as e.g. Microsoft, Apple and Google are integrating LLMs in their applications and operating systems. Hence, the question on how to make the systems more robust against jailbreaking is a very important question.

This paper offers a statistical approach to this question and presents experiments. The mathematics is based PAC-Bayesian approach and builds upon a very nice formalization of splitting a prompt into queries and concepts.

**Strengths:**

The mathematics are presented very well. In particular, the intuitions and interpretations of the mathematical concepts are presented well, and can also be followed by readers, who are not able to follow all the details of every equation.

Overall, I really appreciated the formal approach to modelling jailbreaking and the conclusions, that it is impossible to avoid jailbreaking.

**Weaknesses:**

There is a disconnect between the theory and the experiments. The definitions, theorems, etc. are all presented and motivated very nicely, but could also apply to any other mapping of a stochastic system. Oversimplified, one may state that there is a mapping from an input space into an output space, in which the output space can be divided into desired and undesired outputs. The goal is to reduce the space of undesired outputs. To make this a case specifically about LLMs, there needs to some form of conclusions that feed into the experiments. Unfortunately, I don't see this connection. The experiments use a systems prompt that is preprended to the actual prompt. I do not see, how this follows from the theory.

This is really unfortunate, because I would really like to see, how the experiments connect to the theory.

**Questions:**

Could you please point out the connection between experiment and theory that I have seemed to miss.

**Limitations:**

I don't see any negative social impacts. As stated above, I would like to see a stronger connection between experiments and theory.

---

> ### Author Rebuttal · Authors · 2024-08-06
>
> Dear reviewer T1QG,
>
> Thanks for your thorough review! We are pleased to provide further explanations on our E-RLHF proposal, its theoretical foundations, and its specific relation to LLMs.
>
> *1.  The definitions, theorems, etc. are all presented and motivated very nicely, but could also apply to any other mapping of a stochastic system. ...To make this a case specifically about LLMs, there needs to some form of conclusions that feed into the experiments.*
>
> To better address your concerns, we would like to clarify the **goals** we aim to achieve with our constructed framework to make it meaningful and robust. Our framework should:
>
> - Abstract the capability of an LLM to generalize on unseen but plausible prompts. **This property is particularly important as many representative jailbreak methods exploit this ability to circumvent the guardrail of existing LLMs.** This necessitates a clear distinction between "plausible" and "seen" prompts. Additionally, defining the concept of "generalization" is critical, yet inherently complex.
> - Model the capability of an adversary. For jailbreak attempts, formulating an adversary in a precise mathematical manner is challenging due to the discrete and unbounded nature of the operations performed on prompts.
> - Discriminate between harmful and non-harmful prompts. It is essential that statements on jailbreak focus exclusively on harmful prompts. However, the definition of "harmful" itself remains ambiguous and diffucult to mathematically formalize.
>
>
> **Framework**: Based on these points, we first assume each prompt can be decomposed into a query and a concept. This offers us the  opportunity to model generalization by assuming invariance on the concept, thereby allowing us to mathematically describe the capability of the adversary. Reflecting point (1), in Assumption 4.1, we assume that the domain of the LLM output distribution dependents solely on the concept and not the query. This is a crucial, yet we argue, realistic assumption. We argue that this abstraction is **distinctive of LLMs** and is not easily applicable to generic stochastic systems, as not all mappings within such systems will exhibit the property described in point (1). This characteristic further impacts our E-RLHF proposal, which relies on this essential property to enhance safety. Without this property, the performance of those mappings on **unseen harmful prompts** found by the adversary would remain unaffected, while for LLMs the corresponding safety zone will be enlarged. That said, we are eager to explore whether our proposed framework can be effectively applied to other generic stochastic systems. This extension of our research could potentially broaden the applicability and impact of our findings, offering valuable insights into a wider range of systems.
>
> *2. The experiments use a systems prompt that is preprended to the actual prompt. I do not see, how this follows from the theory. Could you please point out the connection between experiment and theory that I have seemed to miss.*
>
> **Specific implementation is driven by our framework and works**: We further want to convince you that the incorporation of a prepended safe prefix represents an intuitive yet effective implementation of our E-RLHF proposal. Theorem 2 elucidates the relationship between the size of the safety zone and the ability of an adversary to successfully compromise the model. This insight directly informs our practical approach to enlarging the safety zone. Our analysis reveals that the prevalent RL fine-tuning objective, especially its KL-term, inadvertently contributes to vulnerabilities due to the unsafe nature of $p_{\textrm{SFT}}(x)$ when $x$ itself is harmful. By substituting harmful concepts in $x$ with safe alternatives, we can effectively mitigate this issue. We have elaborated several variations of its implementation in discussions with reviewer qvRD. In the experimental section of our study, we take a simple, computationally efficient approach by introducing a safe prefix to $x$.
>
> We want to emphasize the significance of our experimental result. We achieved significant safety  improvement across all tasks in HarmBench with only one generic intervention (one safe prompt). The dataset size and harmful diversity coverage is the largest compared to previous papers (we refer to the general response and section A.2, section B.4 and Table 5 of [1] as reference).
>
> We found that our approach already provides an improvement in safety. We think that with better implementations, a larger alignment dataset, and improved optimization techniques (e.g., with PPO), the enhancement in safety could be significantly increased.
>
>
> Please let us know if the above explanations resolve your concern on how we motivated our framework specifically for LLMs, and how our E-RLHF proposal is linked to the theoretical discoveries.

---

> > ### Comment · Reviewer_T1QG · 2024-08-12
> > **Answer**
> >
> > Dear authors,
> >
> > Thank you very much for the lengthly reply to my main concern. The reply validates, that I did understand the main points of the paper and approach. Unfortunately, I don't see any additional arguments, that directly answer my question. I do see the importance of the experiments and results and I also do so see the theory, that you discuss in the paper. I just don't see, how the two are connected.
> >
> > Prompt engineering and system prompts are a well established method in many LLM-applications. They do not follow from your theory as there are other ways to change the safe and unsafe areas.
> >
> > I appreciate the answer, but will remain with my initial assessment of the paper.

---

> > > ### Author Response · Authors · 2024-08-12
> > > **Response to comments**
> > >
> > > Dear reviewer T1QG,
> > >
> > > Thank you for your response! Could you please provide more details about your concerns so that we can address them more effectively? In our rebuttal, regarding the specificity of our framework to LLMs, we clarify how it is uniquely tailored in our assumptions, which are particularly relevant to LLMs. Concerning the connection between theory and experiment, we had hoped to have clarified how the theory allowed us to identify the problems with current RLHF approaches (keeping a too small safety zone), thus leading us to our successful strategy of safe concept substitution through prefixing. In our discussion with reviewer qvRD, we explore several strategies that involve both human intervention and LLM-based support to facilitate the concept substitution as future explorations.
> > >
> > > We welcome any suggestions on how we can further refine our approach to better address your concerns.

---

### Official Review · Reviewer_qvRD · 2024-07-09

**Soundness:** 3
**Presentation:** 3
**Contribution:** 3
**Rating:** 7
**Confidence:** 4

**Summary:**

The paper provides a theoretical insight about LLM jailbreaks using PAC-Bayesian bound for pretraining LLMs. It assumes that there always exists the harmful data in the mixture, and as the model is trained on this mixture, the model will probably produce the responses in harmful zone (it has a specific definition in the main paper). Based on this framework, the authors suggests that the safety zone of the models should be extended, and to this end, they introduce a method called E-RLHF which can expand the safety zone. E-RLHF replaces the harmful prompts x_h into benign prompts x_s, and replace some of the terms in RLHF (and DPO) to make sure that the model keeps in the safety zone. Experimental results also show that it does not sacrifice the general capabilities but can be improved in the safety perspective.

**Strengths:**

- This paper suggests a theoretical insight about the LLM jailbreaks, which were not addressed much in the previous literature on jailbreaking. The theoretical framework is sound and compelling.
- Based on this framework, the paper also suggests a simple training trick that can lead to better safety training.
- Empirical results also show that their idea is working well, shown by Harmbench, AdvBench, and MT-Bench scores.
- The paper provides extensive evaluation results on jailbreaking setups, providing the results from more than 10 attack setups.

**Weaknesses:**

- E-RLHF is an inaugural and simple form of expanding the safety zone of LLM; I think there could be more sophisticated and effective ways, and I hope the authors will address this in the future works.
- Other than that, I think there is no big weakness in the paper, but have some minor comments:
  - Eq (2): it is slightly confusing that in D_KL, the term only have p_LM(x) and p_SFT(x), not p_LM(e|x) and p_SFT(e|x).
  - About writing: I am not familiar with using the term "explanation" -- instead, I think using "response" is more common. At the first glance, it was hard to comprehend the meaning of the term.

**Questions:**

No specific questions about the paper.

**Limitations:**

The authors provided limitations section in the paper.

---

> ### Author Rebuttal · Authors · 2024-08-06
>
> Dear reviewer qvRD,
>
> We sincerely thank you for reading our work in great detail! Here are our responses to your concerns.
>
> *1. E-RLHF is an inaugural and simple form of expanding the safety zone of LLM; I think there could be more sophisticated and effective ways, and I hope the authors will address this in the future works.*
> -  We sincerely appreciate your feedback. This is a fantastic suggestion and we would love to integreate a more detailed discussion on possible avenues for implementation of an alignment strategy based on our theory in the final version of the paper. Some of our thoughts include;
>
> **E-RLHF Formulation.** As outlined in our general response, our proposed E-RLHF is inspired by the realization that the KL-term in the current prevalent alignment method (RLHF) may inadvertently preserve harmful responses when the input prompt $x$ itself is harmful. The RLHF objective aims to align with human preferences (as reflected by the reward term) while maintaining helpfulness (as reflected by the KL-term). To enhance safety while preserving these characteristics, we propose setting $p_{\textrm{SFT}}(\cdot)$ to a safe distribution. We believe that maintaining the mathematical reward-plus-KL formulation is vital, and currently, we do not see a clear pathway for other formulations to achieve these objectives simultaneously.
>
> **E-RLHF Implementation.** We are considering several alternative strategies to refine our implementation of E-RLHF. The first concerns the harmful prompt filtering step. Instead of our current approach, which involves prompting an LLM to assess whether an input prompt is harmful, a more straightforward method might involve sampling responses and labeling the prompt as harmful if the likelihood of a response being harmful surpasses a predefined threshold. Additionally, involving human annotators to manually design, and identify harmful prompts from existing alignment datasets could be beneficial. Furthermore, considering that the determination of whether content is harmful can vary based on different backgrounds and contexts, the filtering process could also be made adaptive to these conditions. This adaptability is particularly crucial if we are cognizant of the diverse applications of LLMs. By tailoring the filtering mechanisms to accommodate various contexts, we can enhance the safety of the responses generated by LLMs, ensuring they are appropriate and considerate across a spectrum of scenarios, which could improve the utility and acceptance in global applications. The second pertains to the safe concept replacement step. Rather than our current method of safe prefixing, one could involve human annotators to rewrite harmful prompts or prompt a LLM to decompose-and-replace harmful prompts. We opted for safe prefixing in our paper due to its simplicity, which helps avoid excessive computational demands and reduces the need for intensive human labor. We believe that with effective prompt engineering (e.g., curate the prompt in a similar fashion as demonstrated in Table 15 in [1]), this second step can be efficiently implemented using an LLM, which we aim to explore in future work.
>
> However, we want to emphasize that our approach despite its simplicity strikes a balance between computational feasibility and the goal of expanding the safety zone. We are delighted to report that this simple intervention performed exceptionally well against state-of-the-art jailbreaking attack benchmarks. This outcome not only underscores the viability of our proposal but also establishes a promising baseline.
>
> We acknowledge that our approach to expanding the safety zone is just one of many potential strategies, and we are excited to see other researchers incorperate our insights into their alignment strategies. With the integration of additional safety alignment data and the implementation of more sophisticated strategies, our method holds the potential to deliver even more impressive results.
>
> We would also be delighted to see our theoretical framework be applied in other domains as suggested by reviewer T1QG.
>
> *2. Eq (2): it is slightly confusing that in D_KL, the term only have $p_{LM}(x)$ and $p_{\textrm{SFT}}(x)$, not $p_{LM}(e|x)$ and $p_{\textrm{SFT}}(e|x)$.*
>
> - We apologize for any confusion caused by our notation. Throughout the paper, we denote the distribution over responses as $p_{LM}(q,c)=p_{LM}(x)$. With this notation, and incorporating a reward model $r(x,e)$, the RL fine-tuning can be expressed either as $\mathbb E_{x\sim\mathcal D_s, e\sim p_{LM}(\cdot|x)}[r(x,e)-\beta\frac{\log p_{LM}(e|x)}{\log p_{\textrm{SFT}}(e|x)}]$, or as $\mathbb E_{x\sim\mathcal D_s}[\mathbb E_{e\sim p_{LM}(\cdot|x)}[r(x,e)]-\beta \mathbb D_{\textrm{KL}}(p_{LM}(x) || p_{\textrm{SFT}}(x))]$. It is important to note that the expectation over $e$ is used for computing the reward $r(x,e)$, while the $\mathbb D_{\textrm{KL}}$ serves to regularize $p_{LM}(x)$, ensuring it does not deviate significantly from $p_{\textrm{SFT}}(x)$. To avoid any further confusion,  we will clarify this point in our final draft by using the second equation.
>
> *3. I am not familiar with using the term "explanation" -- instead, I think using "response" is more common. At the first glance, it was hard to comprehend the meaning of the term*
>
> - We apologize for the confusion. We use the term "explanation" as a counterpart to "concept", based on the empirical observation that, in most jailbreaking attacks currently considered by the community, the adversary seeks instructions or explanations for a single harmful attempt. We appreciate your feedback and will incorporate it by resuming the use of "response" in our final draft to enhance readability and ensure our notation is easier to follow.
>
> We hope these explanations address your concerns. Please let us know if you need further clarification, we would be happy to discuss further.
>
> References
>
> [1] Jailbreaking Black Box Large Language Models in Twenty Queries

---

> > ### Comment · Reviewer_qvRD · 2024-08-12
> >
> > Thank you for your responses, I will keep my rating.

---

> > > ### Author Response · Authors · 2024-08-12
> > > **Response to comments**
> > >
> > > Dear reviewer qvRD,
> > >
> > > We appreciate the acknowledgements of our work and thanks for responding to our rebuttal!

---

### Author Rebuttal · Authors · 2024-08-07

We sincerely thank all reviewers for taking the time to review our paper and providing valuable feedback. We appreciate the recognition of our established theoretical framework and the acknowledgment of the nuanced formulation of our results from all reviewers. We had to traverse several conceptual steps to overcome the challenge that the adversary can modify the input prompt in an unbounded, unconstrained way. Our key proposal on the decomposition of input prompts into the (query, concept) pair has enabled us to formalize the adversary mathematically. Within our framework, we offer a clear distinction between harmful and non-harmful prompts, elucidate the generalization capabilities of LLMs on unseen prompts, and demonstrate the difficulty and impossibility of defending against jailbreak attempts by presenting a novel statistical bound.

The reviewers have raised several important concerns that we are eager to address.

1. The connection between our theory and the proposed experimental strategy (T1QG, UXGw).
2. The simplicity and feasibility of our implementation (qvRD, UXGw).
3. The effectiveness of our E-RLHF proposal (sD4q).

We appreciate these insightful comments and are pleased to provide detailed clarifications to each of these points below.

**Connection of our framework to experiments.** We identified a significant limitation for safety inherent in the widely adopted alignment strategy, RLHF. Our analysis has traced the problem back to the KL term, which inadvertently ensures that even the optimal solution retains all harmful responses in the LLM's output. To address this, we have introduced an innovative modification to the KL term concerning harmful prompts. Our approach involves filtering out harmful prompts and replacing them with safer alternatives. We believe it is the most natural and effective solution to mitigate the identified risk. It is important to emphasize that our E-RLHF algorithm is inspired and fundamentally driven by our theoretical results.

**The safe prefix implementation.** Upon introducing E-RLHF, our next challenge was to devise an effective implementation. While we acknowledge the existence of more sophisticated methods (as discussed with reviwer qvRD), we opted for **a simple approach: appending a safe prefix to the harmful prompts.** We find this simplicity particularly compelling. Remarkably, with our strategy applied with a limited alignment dataset and optimized using DPO, we achieve significant improvements in safety. This opens the door to find more nuanced, improved implementations, which have the potential to improve results further.

**Soundness and effectivenss of our E-RLHF proposal.** We used a **recently released jailbreak benchmark: HarmBench**. HarmBench assesses LLM safety using a suite of the most advanced jailbreak adversaries, and scores safety across diverse harm categories including but not limited to Cybercrime & Unauthorized Intrusion, Chemical & Biological Weapons/Drugs, Copyright Violations, Misinformation & Disinformation, Harassment & Bullying, Illegal Activities, and General Harm. The HarmBench dataset comprises 400 prompts. Additionally, we included the AdvBench first 100 subset, a dataset frequently used in previous research. We include results from both datasets to ensure completeness and robustness in our evaluation. Our method has shown significant improvements on both benchmarks and across all categories of harm without any task specific adaptation.

We would be eager to summarize our contributions again as follows.

- **A new Theoretical Framework.** We build a novel framework to analyze LLM jailbreaking. This framework addresses challenges such as abstracting the LLM's generalization capability on unseen prompts, mathematically defining the adversary, and distinguishing between harmful and non-harmful prompts.
- **Theoretical evidence on difficulty and impossibility of avoiding jailbreak attacks.** Following our nuance construction of the framework, we offer theoretical evidence that highlights the inherent difficulties and impossibilities of completely avoiding LLM jailbreak attacks.
- **Insight into a RLHF Objective Drawback.** We provide a critical insight into the drawback of the current RLHF objective for safety, which exacerbates the problem for post-alignment LLMs.
- **Theory-Inspired Proposed Solution: E-RLHF.** In response to this insight, we propose an algorithmic framework, E-RLHF, designed to address and mitigate this drawback.
- **Effective Implementation and Results.** We implement a **simple yet effective** version of E-RLHF and demonstrate its superior safety performance **across a suite of diverse jailbreak adversaries without task specific adaptations**.


Finally, we want to take the opportunity to emphasize the importance of our work.
- The adaptability of our E-RLHF. Our approach can be tailored to align with various cultural norms and contexts, such as in educational, medical, and legal settings. This customization could be achieved by filtering harmful prompts based on specific backgrounds and applying our safe prompt replacement strategy.
- Defense strategies are crucial for LLM deployment, and our insights underscore the vulnerability to (even) single interaction attacks. We hope this insight will spark further research. If perfect defense is impossible, we need to rethink how applications should be designed acknowledging these limitations. We hope our research can foster a more robust and thoughtful approach to the deployment of LLMs.


We sincerely thank all reviewers for their time on our rebuttal. We hope our response addresses your concerns and highlights the significance of our contributions, both theoretically and experimentally. We would be delighted to discuss further in the next week.

---

> ### Author Response · Authors · 2024-08-13
>
> We want to sincerely thank the reviewers for their effort in thoughtfully discussing our paper. We appreciate your detailed feedback and the positive points you’ve highlighted about our work. We firmly believe that our contributions, particularly in demonstrating the inevitability of jailbreaking, are crucial for fundamentally understanding the limits of alignment. This work hence delineates the critical battleground where future attack and defense strategies.
>
> While we understand your concerns, we kindly request that you consider an increase in the score to better reflect the strengths and potential impact of our work. We are fully committed to further improving the manuscript and remain open to any additional suggestions you may have.
>
> Warm regards + fantastic start in the week,
> The authors

---

### Decision · Program_Chairs · 2024-09-25

**Decision:**

Accept (poster)

**Comment:**

The paper provides a theoretical analysis of jailbreaking for LLMs using PAC-Bayesian bound. They found that the jailbreaking is unpreventable.

Strength:
S1. The paper proposes a sound theoretical framework to analyze the jailbreaking behavior of LLMs.
S2. the paper proposes an improved alignment training method E-RLHF.

Weakness:
W1. There is disconnection between the theoretical analysis and experimental validation.

Overall, the theoretical framework proposed in this paper to analyze LLM jailbreaking is an important contribution.